# Maximizing and Satisficing in Multi-armed Bandits with Graph Information*

**Parth K. Thaker**
Arizona State University
pkthaker@asu.edu

**Mohit Malu**
Arizona State University
mmalu@asu.edu

**Nikhil Rao**
Microsoft
nikhilrao86@gmail.com

**Gautam Dasarathy**
Arizona State University
gautamd@asu.edu

## Abstract

Pure exploration in multi-armed bandits has emerged as an important framework for modeling decision making and search under uncertainty. In modern applications however, one is often faced with a tremendously large number of options and even obtaining one observation per option may be too costly rendering traditional pure exploration algorithms ineffective. Fortunately, one often has access to similarity relationships amongst the options that can be leveraged. In this paper, we consider the pure exploration problem in stochastic multi-armed bandits where the similarities between the arms is captured by a graph and the rewards may be represented as a smooth signal on this graph. In particular, we consider the problem of finding the arm with the maximum reward (i.e., the maximizing problem) or one that has sufficiently high reward (i.e., the satisficing problem) under this model. We propose novel algorithms **GRUB** (GRaph based UcB) and $\zeta$-**GRUB** for these problems and provide theoretical characterization of their performance which specifically elicits the benefit of the graph side information. We also prove a lower bound on the data requirement that shows a large class of problems where these algorithms are near-optimal. We complement our theory with experimental results that show the benefit of capitalizing on such side information.

## 1 Introduction

The multi-armed bandit has emerged as an important paradigm for modeling sequential decision making and learning under uncertainty. Practical applications include design policies for sequential experiments [44], combinatorial online leaning tasks [9], collaborative learning on social media networks [30, 4], latency reduction in cloud systems [23] and many others [8, 59, 50, 24]. In the traditional multi-armed bandit problem, the goal of the agent is to sequentially choose among a set of actions or arms to maximize a desired performance criterion or reward. This objective demands a delicate tradeoff between exploration (of new arms) and exploitation (of promising arms). An important variant of the reward maximization problem is the identification of arms with the highest (or near-highest) expected reward. This *best arm identification* [41, 13] problem, which is one of pure exploration, has a wide range of important applications like identifying and testing drugs to treat infectious diseases like COVID-19, finding relevant users to run targeted ad campaigns, hyperparameter optimization in neural networks and recommendation systems. The broad range of

---
*This work was supported in part by the National Science Foundation through the awards CCF-2048223, CNS-2003111, CCF-2029044, and OAC-1934766. This work was also supported partly by the ASU SenSIP Center

applications of this paradigm is unsurprising given its ability to essentially model any optimization problem of black-box functions on discrete (or discretizable) domains with noisy observations.

While pure exploration problems in bandits show considerable promise, there are significant hurdles to their practical usage. In modern applications, one is often faced with a tremendously large number of options (sometimes in the order of hundreds of millions) that need to be considered for decision making. In such cases, playing (i.e., obtaining a random sample from) each bandit arm even once could be intractable. This renders traditional approaches to pure exploration ineffective. Fortunately, in several applications, the arms and their rewards are related to each other and information about the reward of one arm may be deduced from plays of similar arms. In this paper, we consider the pure exploration problem in stochastic multi-armed bandits where the similarities between arms is captured by a graph and the rewards may be represented as a smooth signal on this graph. Such graph side information is available in a wide range of applications: search and recommendation systems have graphs that capture similarities between items [17, 43, 53, 11]; drugs, molecules and their interactions can be represented on a graph [19]; targeted advertising considers users connected to each other in a social network [20], and hyperparameters for training neural network are often inter-related [57]. It is worth noting that such graphs are sometimes intrinsic to the problem (e.g., spatial coordinates or social/computer networks), or may be inferred based on a similarity metrics defined on arm features; a recent line of work considers constructing such graphs to enable more effective learning [see e.g., 58, 31].

**Our Contributions:** We consider the pure exploration in multi-arm bandits problem when a graph that captures similarities between the arms is available. In particular, we consider the problem of finding the arm with the maximum reward (i.e., the maximizing problem) or one that has sufficiently high reward (i.e., the satisficing problem[2]) under the assumption that arm rewards are smooth with respect to a known graph. Our main contributions may be summarized as follows:

**(a)** We devise a novel algorithm GRUB for the best arm identification problem (i.e., the maximizing problem) that specifically exploits the *homophily* (strong connections imply similar average rewards) on the graph (Section 3).

**(b)** We provide a theoretical characterization of the performance of GRUB. To this end, we define a novel measure $\mathfrak{I}$ that we dub the "*influence factor*" which depends on the resistance distance of the underlying graph. This measure captures the benefit of the graph side information and plays a central role in the analysis of GRUB. In the traditional (graph-free) best arm identification problem, the sample complexity is know to scale as $\sum_{i=1}^{n} \frac{1}{\Delta_i^2}$, where $\Delta_i$ is the gap between the expected rewards of the best arm and arm $i$. On the other hand, we show that GRUB roughly has a complexity that scales like $\sum_{i \in \mathcal{H}} \frac{1}{\Delta_i^2}$ samples where the set $\mathcal{H}$ is a set dependent on the influence factor, which contains arms which are hard to distinguish from optimal arm. For a broad range of problems $|\mathcal{H}| \ll n$, yielding significant improvement over traditional best arm identification algorithms (Section 4).

**(c)** In Section 5, we provide lower bounds on the minimum number of samples required for identification of the optimal arm when a graph encoding arm similarities is available. This shows the near-optimality of GRUB for an important class of representative problems.

**(d)** In many real world scenarios, the aim of finding the absolute best arm can often be too costly or even intractable. In these situations, it may be more appropriate to solve the *satisficing* problem, where the algorithm returns an arm that is good enough. We propose a variant of GRUB, dubbed $\zeta$-GRUB for this important setting in Section 6

**(e)** Finally, in Section 7, we complement our theoretical results with an empirical evaluation of our algorithms. We further provide algorithmic improvements to GRUB and discuss novel sampling policies for best arm identification in the presence of graph information.

## 1.1 Related Work

The textbook [32] is an excellent resource for the general problem of multi-armed bandits. The pure exploration variant of the bandit problem is more recent, and has also received considerable attention in the literature [6, 7, 15, 14, 4, 21]. These lines of work treat the bandit arms or actions as independent entities and playing a particular arm yields no information about any other arm. This leads to great difficulty in scaling such methods, since in the problem setups with large number of

---

[2]named after Herbert Simon's celebrated alternative model of decision making [48]

arms, attempting to play *all* arms is not practical. We resolve this precise roadblock by introducing a convenient way of of appending graph side information into the mix which provably accelerates the process of sub-optimal arm elimination (potentially without playing it even once!)

A recent line of work [35, 32, 18, 56, 16, 39] has proposed the leveraging of structural side-information for the multi-armed bandit problem for regret minimization. Such topology-based bandit methods work under the assumption that pulling an arm reveals information about other, correlated arms [18, 47], which help in developing better regret methods. Similarly, spectral bandits [29, 56, 51] assume user features are modelled as signals defined on an underlying graph, and use this to assist in learning. The works [3] and [54] consider similar graph information models, albiet at a degraded level. The authors in [33] use the graphs to improve the regret bounds in a thresholding bandit setting. Work revolving around spectral bandits utilize the *spectrum* of the graph laplacian. In contrast, we focus on the *combinatorial properties* of the graphs to devise algorithms and analyse them. Another line of work [12, 52, 36, 37] considers search problems on graphs under a different model and there is an opportunity for future work to combine these techniques.

Most of the aforementioned works focus on regret minimization in the presence of graph information. The problem of pure exploration with similarity graphs has received far less attention. The authors in [29] were the first to attempt at filling this gap for the spectral bandit setting. They provide an information-theoretic lower bound and a gradient-based algorithm to estimate this lower bound to sample the arms. The authors provide performance guarantees for the algorithm, but these results only indirectly capture the benefit brought by the graph; our results on the other hand are based on a novel complexity measure that explicitly elicits the benefit of having the graph side information.

Note that, similarity graph information considered in this work is fundamentally different from linear rewards assumption in contextual/linear bandits. In the linear bandits problem, the reward behavior is assumed to be low dimensional and this is crucial for the improved regret bounds and sample complexity guarantees [32, 49]. In the current work we do not make any assumptions on low dimensionality of the rewards but still show improvements in sample complexity provided a good arm-similarity graph is available. We show a toy example in Appendix H where a low dimensional linear bandit cannot be competitive with the corresponding graph-bandit setting.

## 2   Problem Setup and Notation

We consider an $n$-armed bandit problem with the set of arms given by $[n] \triangleq \{1, 2, 3, \ldots, n\}$. Each arm $i \in [n]$ is associated with a $\sigma$-sub-Gaussian distribution $\nu_i$. That is, $\mathbb{E}_{X \sim \nu_i} [\exp(s(X - \mu_i))] \le \exp\left(\frac{\sigma^2 s^2}{2}\right) \forall s \in \mathbb{R}$, where $\mu_i = \mathbb{E}_{\nu_i} [X]$ is said to be the (expected or mean) reward associated to arm $i$. We will let $\boldsymbol{\mu} \in \mathbb{R}^n$ denote the vector of all the arm rewards. A "play" of an arm $i$ is simply an observation of an independent sample from $\nu_i$; this can be thought of as a noisy observation of the corresponding mean $\mu_i$. The goal of the best-arm identification problem is to identify, from such noisy samples, the arm $a^* \triangleq \arg\max_{i \in [n]} \mu_i$ that has the maximum expected reward, denoted by $\mu^*$. For each arm $i \in [n]$, we will let $\Delta_i \triangleq \mu^* - \mu_i$ denote the sub-optimality of the arm.

As discussed in Section 1, our goal is to consider the best-arm identification where one has additional access to information about the similarity of the arms under consideration. In particular, we model this side information as a weighted undirected graph $G = (V_G, E_G, A_G)$ where the vertex set, $V_G = [n]$, is identified with the set of arms, the edge set $E_G \subseteq \binom{[n]}{2}$, and adjacency matrix $A_G \in \mathbb{R}^{n \times n}$ describes the weights of the edges $E$ between the arms which captures the similarity in means of connected arms; the higher the weight, the more similar the rewards from the corresponding arms. We will let $L_G = D_G - A_G$ denote the combinatorial Laplacian[3] of the graph [10], where $D_G = \text{diag}(A_G \times \mathbb{1}_n)$ is a diagonal matrix containing the weighted degrees of the vertices. We will suppress the dependence on $G$ when the context is clear. Subsequently, we show that if one has access to this graph and the vector of rewards $\boldsymbol{\mu}$ is *smooth* with respect to the graph (that is, highly similar arms have highly similar rewards), then one can solve the pure exploration problem extremely efficiently. We will capture the degree of smoothness of $\boldsymbol{\mu}$ with respect to the graph using

---

[3]All our results continue to hold if this is replaced with the normalized, random walk, or generalized Laplacian.

the following seminorm[4]:

$$\|\boldsymbol{\mu}\|_G^2 \triangleq \langle \boldsymbol{\mu}, L_G \boldsymbol{\mu} \rangle = \sum_{\{i,j\} \in E_G} A_{ij}(\mu_i - \mu_j)^2. \qquad (1)$$

The second equality above can be verified by a straightforward calculation. Also, notice that $\|\boldsymbol{\mu}\|_G$ being small implies $\mu_i \approx \mu_j$ for $(i,j) \in E$. In such scenario we say that the mean vector $\boldsymbol{\mu}$ is smooth over graph $G$. This observation has inspired the use of the Laplacian in several lines of work to enforce smoothness on the vertex-valued functions [2, 51, 60, 33]. For $\epsilon > 0$, we say that arms (rewards) are $\epsilon$-smooth with respect to a graph $G$ if $\|\boldsymbol{\mu}\|_G \leq \epsilon$.

Let $\mathcal{C}(G) \subset 2^{[n]}$ denote the set of all connected components and let $k(G) \triangleq |\mathcal{C}(G)|$ denote the number of connected components of the graph $G$. For a vertex $i \in [n]$, we will let $C_i(G) \in \mathcal{C}(G)$ denote the connected component that contains $i$. When the context is clear we sometimes let $C_i(G)$ also refer all the nodes in the connected component. We say a graph $G = ([n], E)$ has $k$-*isolated cliques* if it can be divided into fully connected sub-graphs $G_i = (V_i, E_i)$ such that $V_i \subseteq [n], E_i = \binom{V_i}{2}$ for all $i \in [k], V_i \cap V_j = \emptyset, E_i \cap E_j = \emptyset$ for all $i, j \in [k]$, and $\bigcup_{i=1}^k V_i = [n], \bigcup_{i=1}^k E_i = E$. Notice that we only have one clique if $G$ is fully connected.

To solve the best-arm identification problem, we need a sampling policy to sequentially and interactively select the next arm to play, and a stopping criterion. For any time $t \in \mathbb{N}$, the sampling policy $\boldsymbol{\pi}_t = \{\pi_s\}_{s \leq t}$ is a function that maps $t$ to an arm in $[n]$ given the history of observations up to time $t - 1$. With slight abuse of notation, we will let $\pi_t$ denote the arm chosen by an agent at time $t$. Let $r_{t,\pi_t}$ denote the random reward observed at time $t$ from arm $\pi_t$. We use $t_i(\boldsymbol{\pi}_t)$ (referred as $t_i$ for simplicity) to denote the number of times arm $i$ is played under the sampling policy $\boldsymbol{\pi}_t$. In this paper we tackle the following problems:

**P1 (Best arm identification):** *Given $n$ arms and an arbitrary graph $G$ capturing similarity between the arms, can we design a policy $\boldsymbol{\pi}_T$ that exploits the similarity to find the best arm efficiently?*

**P2 ($\zeta$-best arm identification):** *Under the setting in **P1**, can we design a similarity exploiting policy $\boldsymbol{\pi}_T$ so as to find an arm belonging to the set $B(\zeta) \triangleq \{i \in [n] : |\mu_i - \mu_{a^*}| \leq \zeta\}$ efficiently?*

## 3 The GRUB Algorithm

We now introduce GRUB (GRaph based Upper Confidence Bound), a novel but natural algorithm for best arm identification in the presence of graph side information. We begin with an intuitive description of how GRUB incorporates the graph side information into an *upper confidence bound* (UCB) strategy. Most UCB algorithms [32, 51] compute the estimates of mean and variance, and use these to eliminate arms that have been deduced to be sub-optima. The key idea behind GRUB is that the arm similarity information allows us to create high-quality estimates of mean rewards and confidence intervals for arms that have not been (sufficiently) sampled yet. In what follows, we describe the building blocks of GRUB.

### 3.1 Leveraging Graph Side Information

We introduce two key ideas that lie at the heart of the GRUB algorithm. First, at each step, GRUB computes a regularized estimate of the means of *all the arms*; the regularization based on the graph Laplacian essentially promotes the smoothness of the mean vector on the given graph. This allows the algorithm to estimate means of arms it has *never sampled*. To do this, at any given time step $T$, the algorithm solves the following Laplacian-regularized least-squares optimization program:

$$\hat{\boldsymbol{\mu}}_T = \arg\min_{\boldsymbol{\mu} \in \mathbb{R}^n} \left\{ \left[ \sum_{t=1}^T (r_{t,\pi_t} - \mu_{\pi_t})^2 \right] + \rho \langle \boldsymbol{\mu}, L_G \boldsymbol{\mu} \rangle \right\}, \qquad (2)$$

---

[4] $L_G$ is not positive definite, and can be verified to have as many zero eigenvalues as the number of connected components in $G$

where $\rho > 0$ is a tunable parameter. Equation (2) admits a closed form solution of the form

$$\hat{\boldsymbol{\mu}}_T = \left( \sum_{t=1}^{T} \mathbf{e}_{\pi_t} \mathbf{e}_{\pi_t}^{\top} + \rho L_G \right)^{-1} \left( \sum_{t=1}^{T} \mathbf{e}_{\pi_t} r_{t,\pi_t} \right),$$

provided the matrix $V_T \triangleq \sum_{t=1}^{T} \mathbf{e}_{\pi_t} \mathbf{e}_{\pi_t}^{\top} + \rho L_G$ is invertible; $\mathbf{e}_i$ denotes the $i$-th standard basis vector for the Euclidean space $\mathbb{R}^n$. In Appendix B we show that invertibility holds if and only if the sampling policy yields at least one sample per connected component of $G$. This is a rather mild condition that we arrange for explicitly in our algorithm, given that we know the graph $G$. In what follows we assume that every connected component of graph $G$ is sampled at least once. This regularized mean estimation procedure yields an estimate of the mean that is both in agreement with observations and smooth on the graph – thereby allowing information sharing among similar arms.

The second key idea of our algorithm is the utilization of the graph $G$ in tracking the confidence bounds of *all the arms simultaneously*. Intuitively, for identifying the best arm, we must be reasonably certain about the sub-optimality of the other arms. This in turn would require the algorithm to track a high-probability confidence bound on the means of all the arms. In the traditional (graph-free) best arm identification problem, the confidence interval of an arm's mean estimate depends on the number of times the arm has been played. Requiring multiple plays of all suboptimal arms for obtaining high confidence bounds is potentially disastrous when the number of arms is very large. In our setup, we show that the knowledge of the similarity graph greatly improves this situation. In particular, we show that a play of any arm not only tightens its own confidence interval, but also has an impact on the confidence intervals of *all connected arms*. To quantify the benefit of graph information for the confidence bounds, we will define a novel quantity for each arm – the effective number of plays.

**Definition 3.1** (Effective Number of Plays). Let $\rho > 0$ and $\{t_i\}_{i=1}^{n}$ denote the number of plays of each of the $n$ arms when a sampling policy $\boldsymbol{\pi}_T$ is employed for $T$ time steps. Suppose that for each connected component $C \in \mathcal{C}(G)$, there is at least one arm $i_C \in C$ such that $t_{i_C} > 0$. Then the effective number of plays for each arm $i \in [n]$ is defined as $t_{\text{eff},i} \triangleq \left[ (N_T + \rho L_G)^{-1} \right]_{ii}^{-1}$, where $N_T$ is a diagonal matrix of $\{t_i\}_{i=1}^{n}$, and $L_G$ denotes the Laplacian of the given graph $G$.

Effective number of plays $t_{\text{eff},i}$ for any arm $i$ is influenced by two factors: (a) the number of samples of arm $i$ itself, and (b) the number of samples of any arm in the connected component $j \in C(i), j \neq i$. It can be shown that for any arm $i$, $t_{\text{eff},i}$ depends on the number of connections of node $i$ in graph $G$ and its value increases as the connectivity of the node increases. The choice of the terminology for this quantity is justified by the following lemma, which provides a high confidence bound for the mean estimate of each arm .

**Lemma 3.2** (Concentration inequality). *For any $T > k(G)$, the following holds with probability at least $1 - \delta$:*

$$|\hat{\mu}_T^i - \mu_i| \leq \sqrt{\frac{1}{t_{\text{eff},i}}} \left( 2\sigma \sqrt{14 \log \left( \frac{2w_i(\boldsymbol{\pi}_T)}{\delta} \right)} + \rho \|\boldsymbol{\mu}\|_G \right), \quad \forall i \in [n] \tag{3}$$

*where $w_i(\boldsymbol{\pi}_T) = a_0 n t_{\text{eff},i}^2$ for any constant $a_0 > 0$, $\hat{\mu}_T^i$ is the $i$-th coordinate of the estimate from (2)*

Notice that the *effective number of plays* has a similar role as the number of plays in traditional pure exploration algorithms [13]. Indeed, in the absence of graph information, $t_{\text{eff},i}$ reduces to $t_i$, the total number of plays of individual arms. Lemma 3.2 recovers high confidence bounds for standard best-arm identification problem [13]. It should be noted that while our work is the first to identify this interpretable quantity explicitly, the result of Lemma 3.2 in other forms has appeared before in the literature [1, 51, 56].

We introduce our algorithm GRUB for best arm identification when the arms can be approximately cast as nodes on a graph. GRUB uses insights from graph-based mean estimation (2) and upper confidence bound estimation (3) for its elimination policies to search for the optimal arm.

GRUB accepts as input a graph $G$ on $n$ arms (and its Laplacian $L_G$), a regularization parameter $\rho > 0$, a smoothness parameter $\epsilon > 0$, and an error tolerance parameter $\delta \in (0, 1)$. It is composed of the following major blocks.

**Initialization:** First, GRUB identifies the clusters in the $G$ using a `Cluster-Identification`

routine. Any algorithm that can efficiently partition a graph can be used here, e.g METIS [25]. GRUB then samples one arm from each cluster. This ensures $V_T \succ 0$, which enables GRUB to estimate $\hat{\boldsymbol{\mu}}_T$ using the closed form solution of eq. (2). A great advantage of GRUB is that the initialization phase only requires steps equal to the number of disconnected components in the graph. This is in direct contrast with traditional best arm identification algorithms, which require atleast one sample from every arm initially.

**Sampling policy:** At each round, GRUB obtains a sample from the arm returned by the routine `Sampling-Policy`, which cyclically samples arms from different clusters while ensuring that no arm is resampled before all arms in consideration have the same number of samples. This is distinct from standard cyclic sampling policies that is traditionally used for best arm identification [13], but any of them may be modified readily to provide a cluster-aware sampling policy for GRUB. In our experiments, we show that replacing cyclic sampling with more statistics- and structure-aware sampling greatly improves performance; a theoretical analysis of these is a promising avenue for future work. One of the major advantage of GRUB is the lite nature of the computation. Every loop just requires a rank-1 inverse update which can be performed very efficiently and it does not need any subroutines, unlike [29]

**Bad arm elimination :** At any time $t$, let $A$ be the set of all arms in consideration for being optimal. Using the uncertainty bound from (3), GRUB uses the following criteria for sub-optimal arm elimination. At each iteration, GRUB identifies an arm $a_{\max} \in A$, $a_{\max} = \arg\max\limits_{i \in A} \left[ \hat{\mu}_t^i - \beta_i(t) \sqrt{t_{\text{eff},i}^{-1}} \right]$, where $\beta_i(t) = \left( 2\sigma \sqrt{14 \log \left( \frac{2na_0 t_{\text{eff},i}^2}{\delta} \right)} + \rho\epsilon \right)$, with the *highest lower bound* on its mean estimate. Following this, GRUB removes arms from the set $A$ according to the following elimination policy,

$$A \leftarrow \left\{ \mathbf{a} \in A \mid \hat{\mu}_t^{a_{\max}} - \hat{\mu}_t^a \leq \beta_a(t) \sqrt{t_{\text{eff},a}^{-1}} + \beta_{a_{\max}}(t) \sqrt{t_{\text{eff},a_{\max}}^{-1}} \right\}. \tag{4}$$

Note that GRUB does not require any optimization innerloop as in [29]. This potentially provides GRUB with a significant computation advantage, especially when the dimensionality of the problem is very large. The pseudocode for GRUB can be found in Appendix E.

Next, we derive performance guarantees on the sample complexity for GRUB to return the best arm with high probability.

## 4   Theoretical Analysis of GRUB

In this section we provide a formal statement of the sample complexity of GRUB. To do this, we first introduce a novel quantity we call *influence factor*. The influence factor of an arm is derived from resistance distance, a classical graph theoretic concept. This adds to the interpretability and understanding of the instances where using graph side information might be of tremendous use to the application. The usage of graph through the influence factor allows us to identify arms that can be eliminated quickly from consideration.

### 4.1   Resistance Distance and Influence Factor

We first recall the definition of resistance distance in a graph.

**Definition 4.1** (Resistance Distance). [5] For any graph $G$ with $n$ nodes, given a constant $\delta > 0$, the **resistance distance** $r_{\delta,G}(i,j)$ between two nodes $i, j$ is defined as,

$$r_{\delta,G}(i,j) = R_{ii} + R_{jj} - R_{ij} - R_{ji}, \tag{5}$$

where $R \triangleq \left( L_G + \delta \mathbb{1}\mathbb{1}^T \right)^\dagger$; $\dagger$ denotes the Moore-Penrose inverse, $L_G$ is the Laplacian of graph $G$, and $\mathbb{1} \in \mathbb{R}^n$ is the vector of all 1's.

When the context is clear we denote the resistance distance simply as $r_G(\cdot, \cdot)$. The terminology comes from circuit theory: Suppose that an graph $G = ([n], E)$ is thought of as a resistor network on the nodes $[n]$ where each edge $\{i, j\}$ has a unit resistance. Then, the effective resistance between two nodes $i$ and $j$ is precisely the resistance distance $r(i, j)$. It can be shown in general that nodes

that are close by or connected by several paths have a small resistance distance. Given its ability to capture closeness of nodes in graph, the resistance distance has found a broad range of applications and has been the subject of much study; see e.g., [28, 5, 55].

Using the notion of resistance distance, we define the influence factor $\mathfrak{I}(\cdot, G)$ of a vertex below. This novel measure quantifies the impact of the graph on the parameter estimation of arm $j$, and in particular, allows us to use the combinatorial properties of the graph and the arm means to classify arms into two sets: competitive and non-competitive; the definition of these sets follows right after. As our theory will show, the competitive arms are sampled as though we were in the traditional graph-free setting; on the other hand, non-competitive arms are eliminated rapidly, often with zero plays! Indeed, the smoother the reward vector is with respect to the graph, the fewer competitive arms there are – it is this phenomenon that is captured using the influence factor.

**Definition 4.2** (Influence Factor). Let $G$ be a graph on the vertex set $[n]$. For each $j \in [n]$, define **influence factor** $\mathfrak{I}(j, G)$ as:

$$\mathfrak{I}(j, G) = \begin{cases} \min_{i \in C_j(G), i \neq j} \{r_G(i, j)^{-1}\}, & \text{if } |C_j(G)| > 1, \\ 0, & \text{otherwise .} \end{cases} \tag{6}$$

Here, $r_G(i, j)$ is the resistance distance between arm $i$ and $j$ in $G$ as in Definition 4.1.

**Definition 4.3** (Competitive and Non-Competitive Arms). Fix $\boldsymbol{\mu} \in \mathbb{R}^n$, graph $D$, regularization parameter $\rho$, confidence parameter $\delta$, and smoothness parameter $\epsilon$. We define $\mathcal{H}_D$ to be the set of competitive arms and $\mathcal{N}_D$ to be the set of non-competitive arms as follows:

$$\mathcal{H}_D = \left\{ j \in [n] \Big| \Delta_i \leq 2\sqrt{\frac{2}{\rho \mathfrak{I}(i, D)}} \left( 2\sigma \sqrt{14 \log\left(\frac{2a_0 n \rho^2 \mathfrak{I}(i, D)^2}{\delta}\right)} + \rho\epsilon \right) \right\} \tag{7}$$

and $\mathcal{N}_D \triangleq [n] \setminus \mathcal{H}_D$.

As the name suggests, the arms in $\mathcal{H}$ are close to the optimal arm $a^*$ in mean (competitive compared to the optimal arm $a^*$) and requires several plays before they can be discarded, as shown in the theorem below. Note from the above definition that an arm is more likely to be part of this set if its mean is high (i.e., $\Delta_i$ is low) and its influence factor is low. Similarly, the non-competitive set is composed of arms whose means are not competitive with the optimal arm.

Armed with these definitions, we are now ready to state our main theorem that characterizes the performance of GRUB.

## 4.2 Sampling policy performance

Cyclic sampling policies have been traditionally used in multi-armed bandit problems for best-arm identification [13]. The sample complexity bound for GRUB with cyclic sampling is as follows:

**Theorem 4.4** (GRUB Sample Complexity). *Consider $n$-armed bandit problem with mean vector $\boldsymbol{\mu} \in \mathbb{R}^n$. Let $G = (V, E)$ be the similarity graph with the vertex set $V = [n]$ and edge set $E$, let $\mathcal{G}$ be the set of subgraphs of $G$, and further suppose that $\boldsymbol{\mu}$ is $\epsilon$-smooth i.e., $\|\boldsymbol{\mu}\|_G \leq \epsilon$. Define*

$$T_{\text{sufficient}} \triangleq \arg\min_{D \in \mathcal{G}} \sum_{C \in \mathcal{C}_D} \left[ \sum_{\substack{i \in C \cap \mathcal{H}_D \\ i \neq 1}} \frac{1}{\Delta_i^2} \left[ c_1 \log \frac{c_2}{\delta \Delta_i} + \frac{\rho\epsilon}{2} \right] + \max_{i \in C \cap \mathcal{N}_D} \frac{2}{\Delta_i^2} \left[ c_1 \log \frac{c_2}{\delta \Delta_i} + \frac{\rho\epsilon}{2} \right] \right],$$

*where $\Delta_i = \mu^* - \mu_i$ for all suboptimal arms, $\mathcal{H}_D$ and $\mathcal{N}_D$ are as in Definition 4.3, $\mathcal{C}_D$ is the set of connected components of a given graph $D$ and $c_1, c_2$ are constants independent of system parameters. Then, with probability at least $1 - \delta$, GRUB: (a) terminates in no more than $T_{\text{sufficient}}$ rounds, and (b) returns the best arm $a^* = \arg\max_i \mu_i$.*

*Remark* 4.5. The required number of samples for successful elimination of suboptimal arms, and therefore the successful identification of the best arm, can be split into two categories based on the sets defined in Definition 4.3. Each sub-optimal *highly competitive arm* $j \in \mathcal{H}$ requires $\mathcal{O}(1/\Delta_j^2)$ samples, which is comparable to the classical (graph-free) best-arm identification problem. Additionally, the non-competitive arms $\mathcal{N}$ can be eliminated without being played, depending on the influence factor:

one round of the cyclic sampling suffices to eliminate these arms (even if they are never played!). We refer the reader to Appendix E for a more detailed discussion. Indeed, the smaller $|\mathcal{H}|$ is, the more the graph side information benefits GRUBand vice-versa.

*Remark* 4.6. Note that $T_{\text{sufficient}}$ in Theorem 4.4 involves the minimum over all subgraphs. As we show in Lemma I.8 in the appendix, $\mathfrak{I}$ can actually increase if one restricts their attention to certain subgraphs of $G$; this in turn increases the size of $\mathcal{N}$ and decreases the size of $\mathcal{H}$, hence, giving a tighter upper bound on the performance of the algorithm. GRUB *automatically adapts to the best subgraph* to maximize the influence factor $\mathfrak{I}(\cdot,\cdot)$ to obtain the best possible sample complexity and this is reflected in the statement of Theorem 4.4.

The complete proof of Theorem 4.4 can be found in Appendix E, where we also provide more insights on the behavior of the confidence bound as a function of the number of samples acquired. These results may be of independent interest to the reader.

## 5  Lower Bounds

Let us consider an $n$-armed bandit setup with arm indices $[1, \ldots, n]$. Let $\mu^*$ indicate the mean of the optimal arm and $\mu_i$ indicate the mean values of all other arms such that $\mu_i < \mu^*$. For the rest of this section, without loss of generality, let the index of optimal arm be 1.

**Theorem 5.1.** *Given an $n$-armed bandit model with associated mean vector $\boldsymbol{\mu} \in \mathbb{R}^n$ and similarity graph $G$ smooth on $\boldsymbol{\mu}$, i.e. $\langle \boldsymbol{\mu}, L_G \boldsymbol{\mu} \rangle \leq \epsilon$, for any $0 < \epsilon < \epsilon_0$. Let $G = ([n], E)$ be the graph with only isolated cliques and w.l.o.g let arm 1 be the optimal arm. Then define*

$$T_{necessary} = \sum_{C \in \mathcal{C}_G / C^*} \min_{j \in C} \left\{ \frac{4\sigma^2 \log 5}{(\Delta_j - \sqrt{\epsilon})^2} \right\} + \sum_{j \in C^*/1} \frac{4\sigma^2 \log 5}{\Delta_j^2}, \tag{8}$$

*where $C^*$ is the clique with the optimal arm and $\epsilon_0 := \min\limits_{i \in [n]/1, j \in C(i)} \left[ \Delta_j \left[ 1 - \frac{\Delta_i}{\sqrt{\Delta_i^2 + \Delta_j^2}} \right] \right]^2$. Then any $\delta$-PAC algorithm will need at-least $T_{necessary}$ steps to terminate, provided $\delta \leq 0.1$.*

Using Theorem 5.1, we can show that GRUB is minimax optimal for a $n$-armed bandit problems for certain class of similarity graph $G$. The following result shows that the upperbound on the sample complexity provided in Theorem 4.4 matches the lower bound established in Theorem 5.1 in $\Delta_i$ up to a constant factor.

**Corollary 5.2** (Isolated clusters). *Consider the setup as in Theorem 5.1 with the further restriction that graph $G$ be such that the optimal node is isolated and $\epsilon < \min_{j \in [n]} \frac{\Delta_j^2}{2}$. Define,*

$$T_{necessary} \geq \sum_{C \in \mathcal{C}_G / \{1\}} \max_{j \in C} \left\{ \frac{8\sigma^2 \log 5}{\Delta_j^2} \right\}. \tag{9}$$

*Then any algorithm that takes fewer than $T_{necessary}$ samples will have a probability of error at least 0.1.*

As can be seen in Corollary 5.2, the lower bound expression can scale as standard $n$-armed bandit (implying no added advantage of having graph side-information) or can behave as a $|\mathcal{C}_G|$-armed bandit problem (scales as the number of clusters in graph $G$ rather than number of nodes $n$) purely by changing the similarity graph $G$. The difference between $\mathcal{C}_F$ (connected components in the subgraph constructed by making optimal arm isolated) and $\mathcal{C}_G$ (connected components in the given similarity graph) can lead to more interesting behaviour in terms of lower bound expressions on sample complexity.

## 6  $\zeta$-best-arm identification

It can be observed from Theorem 4.4 that the fact that the means are $\epsilon$-smooth implies that distinguishing arm $j$ from $a^*$ would require at least $O(\epsilon^{-2})$ samples. A tighter upper bound on the violation $\epsilon$ and an edge between $j$ and $a^*$ would make the suboptimal arm $j$ harder to eliminate. However, it stands to reason that in such situations, it might be more practical to not demand for the absolute best

arm, but rather an arm that is nearly optimal. Indeed, in several modern applications we discuss in Section 1, finding an approximate best arm is tantamount to solving the problem. In such cases, a simple modification of GRUB can be used to quickly eliminate definitely suboptimal arms, and then output an arm that is guaranteed to be nearly optimal. To formalize this, we consider the $\zeta$-best arm identification problem as follows.

**Definition 6.1.** For a given $\zeta > 0$, arm $i$ is called $\zeta$-best arm if $\mu_i \geq \mu_{a^*} - \zeta$, where $a^* = \arg\max_i \mu_i$

The goal of the $\zeta$-best arm identification problem is to return an arm $\tilde{a}$ that is $\zeta-$optimal. We achieve this by a simple modification to GRUB, which we dub $\zeta-$GRUB, which ensures that all the remaining arms $i$ satisfy $4\beta(t_i)\sqrt{t_{\text{eff},i}^{-1}} \leq \zeta$. It then outputs the best arm amongst those that are remaining. The following theorem characterizes the sample complexity for $\zeta$-GRUB:

**Theorem 6.2.** *Consider $n$-armed bandit problem with mean vector $\boldsymbol{\mu} \in \mathbb{R}^n$. Let $G$ be the given similarity graph on vertex set $[n]$, and further suppose that $\boldsymbol{\mu}$ is $\epsilon$-smooth. Let $\mathcal{C}$ be the set of connected components of $G$. Define,*

$$T_{\text{sufficient}} \triangleq \arg\min_{D \in \mathcal{G}} \sum_{C \in \mathcal{C}_D} \left[ \sum_{i \in C \cap \mathcal{H}_D} \frac{1}{(\Delta_i \vee \zeta)^2} \left[ c_1 \log \frac{c_2}{\delta(\Delta_i \vee \zeta)} + \frac{\rho\epsilon}{2} \right] \right.$$
$$\left. + \max_{i \in C \cap \mathcal{N}_D} \left\{ \frac{2}{(\Delta_i \vee \zeta)^2} \left[ c_1 \log \frac{c_2}{\delta(\Delta_i \vee \zeta)} + \frac{\rho\epsilon}{2} \right] \right\} \right], \tag{10}$$

*where $\Delta_i = \mu^* - \mu_i$ for all suboptimal arms, $\mathcal{H}_D$ and $\mathcal{N}_D$ are as in Definition 4.3, $\mathcal{C}_D$ is the set of connected components of a given graph $D$ and $\Delta_i \vee \zeta = \max\{\zeta, \Delta_i\}$ and $c_1, c_2$ are constants independent of system parameters. Then, with probability at least $1 - \delta$, $\zeta$-GRUB: (a) terminates in no more than $T_{\text{sufficient}}$ rounds, and (b) returns a $\zeta$-best arm.*

The pseudocode for the $\zeta$-GRUB can be found in Appendix G.

## 7 Experiments

For all our experiments, we use Intel® Core™ i7-10875H CPU @ 2.30GHz × 16 with 32 GB memory. We set $\delta = 1e - 3, \rho = 2.0, \sigma = 2.0$. We evaluate GRUB with different sampling strategies from section J and compare its performance to standard UCB algorithm on both synthetic and real datasets.

**Better sampling strategies:** Theorem 4.4 established a baseline w.r.t. sampling protocol by solving $T_{\text{sufficient}}$ for naive cyclic sampling policy (a sampling policy which does not exploit the graph properties). Note that, even if the sampling policy does not utilize any graph properties, the similarity graph is still being utilized in computing the mean estimate and the confidence widths. For the safe elimination of suboptimal arms, the ultimate goal of GRUB is to shrink the confidence bounds $\beta_i \sqrt{(t_{\text{eff},i})^{-1}}$ as quickly as possible. For the complete description of all the alternatives please refer to Appendix J

**Synthetic Data:** We consider an $n$-armed bandit setup with the aim of finding the best arm. The number of arms scale from $n = 50$ to $200$ in steps of $50$. We consider 2 cases: $G$ is a Stochastic Block model(SBM) with parameters $(p, q) = (0.9, 1e^{-4})$ and $G$ is a Barabási–Albert(BA) graph with parameter $m = 2$, both containing 10 clusters. We run every setup for 20 runs and record the stopping time for all runs. In Figure 1, we compar the baseline cyclic algorithm (Nograph-UCB) with GRUB and its variants (GRUB-MVM, JVM-O, JVM-N), more details on this in Appendix J.

As can be seen in Figure 1, all graph-aware algorithms (GRUB with different sampling policies) outperform the standard UCB based best-arm identification algorithm. Within the different GRUB, different sampling policies exploit the graph infromation in different ways, leading to variations in their performance. GRUB (cyclic sampling based) is outperformed by all other sampling based GRUB methods.

We show additional experiments with different graph parameters for Stochastic block model and Barabási-Albert graphs and different cluster sizes as well as real data in Appendix K. The full code used for conducting experiments can be found at the following Github repository. Discussion about limitations, future works and broader impact are provided in Appendix A.

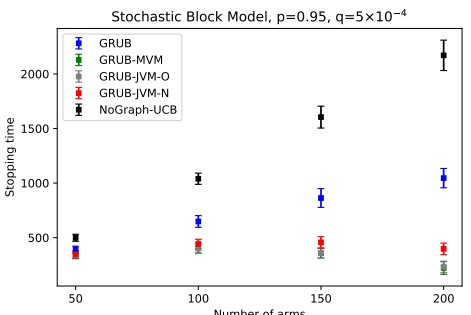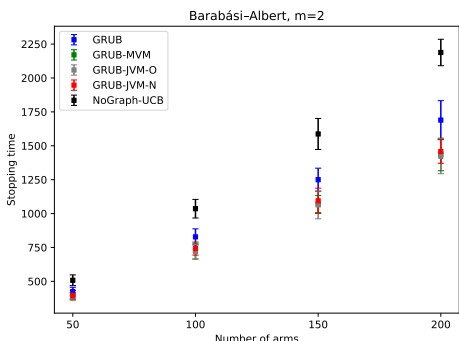

Figure 1: (Best seen in color) Performance of GRUB and its variant sampling protocols for SBM $((p,q) = (0.95, 1e^{-4}))$ [Left] and BA $(m = 2)$ [Right]. GRUB outperforms the standard cyclic UCB method

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
