# Appendix

The appendix is organized as follows. In Appendix A we provide a discussion of the results of the paper, future work, and broader impacts. Appendices B-D and Appendix I provide various supporting results and insights into our main theoretical results. Appendix E and Appendix G provide sample complexity guarantees for GRUB and $\zeta$-GRUB respectively. Appendix F states and proves necessary conditions on the sample complexity, and Appendix H presents a discussion on the incomparability of our graph bandits problem with that of linear bandits. Finally, Appendix J and K contain better sampling strategies and additional experiments respectively.

An anonymized repository containing the code that supports the algorithmic and experimental results of this paper may be found here (see also Appendix L):

## A   Discussion and Broader Impacts

In this work, we consider the problem of best arm identification (and approximate best arm identification) when one has access to information about the similarity between the arms in the form of a graph. We propose a novel algorithm GRUB for this important family of problems and establish sample complexity guarantees for the same. In particular, our theory explicitly demonstrated that benefit of this side information (in terms of the properties of the graph) in quickly locating the best or approximate best arms. We support these theoretical findings with experimental results in both simulated and real settings.

**Future Work and Limitations.**   We outline several sampling policies inspired by our theory in Section 7; an extension of our theoretical results to account for these improved sampling policies is a natural candidate for further exploration. The algorithms and theory of this paper assume knowledge of (an upper bound) on the smoothness of the reward vector with respect to the graph. While this is where one uses domain expertise, this could be hard to estimate in certain real world problems. A generalization of the algorithmic and theoretical framework proposed here that is *adaptive* to the unknown graph-smoothness is an exciting avenue for future work[5,6]. The sub-Gaussianity assumption of this work can also be generalized to other tail behaviors in follow up work. Another limitation of this work is that the statistical benefit of the graph-based quadratic penalization comes at a computational cost – each mean estimation step involves the inversion of an $n \times n$ matrix which has a complexity of $O(n^2 \log(n))$. However, an exciting recent line of work suggests that this matrix inversion can be made significantly faster when coupled with a spectral sparsification of the graph $G$[7,8] while controlling the statistical impact of such a modification. In the context of this problem, this suggests a compelling avenue for future work that studies the statstics-vs-computation tradeoffs in using graph side information.

For this work, we demonstrated the advantages of this side information in pure exploration problems, given knowledge of such an $\epsilon$. Extensions that consider goodness-of-fit and misspecification with respect to the graph $G$ and smoothness parameters $\epsilon$ are interesting avenues for follow up work. Finally, we focus on the ridge-type regularizer of the form $\langle \mu, L_G \mu \rangle$. For future work, it may be productive to expand to a much broader class of regularizers such as those of the form of $\|A\mu\|_q^p$, where $A$ represents a information/ structural constraint matrix and $p, q$ are some positive numbers.

**Potential Negative Social Impacts.**   Our methods can be used for various applications such as drug discovery, advertising, and recommendation systems. In scientifically and medically critical applications, the design of the reward function becomes vital as this can have a significant impact on the output of the algorithm. One must take appropriate measures to ensure a fair and transparent outcome for various downstream stakeholders. With respect to applications in recommendation and targeted advertising systems, it is becoming increasingly evident that such systems may exacerbate

---

[5]T. Tony Cai, Ming Yuan "Adaptive covariance matrix estimation through block thresholding," The Annals of Statistics, Ann. Statist. 40(4), 2014-2042, (August 2012)

[6]Banerjee, T., Mukherjee, G., & Sun, W. (2020). Adaptive sparse estimation with side information. Journal of the American Statistical Association, 115(532), 2053-2067.

[7]Spielman, D. A., & Teng, S. H. (2011). Spectral sparsification of graphs. SIAM Journal on Computing, 40(4), 981-1025.

[8]Vishnoi, Nisheeth K. "Lx= b." Foundations and Trends in Theoretical Computer Science 8.1–2 (2013): 1-141.

polarization and the creation of filter-bubbles. Especially the techniques proposed in this paper could reinforce emerging polarization (which would correspond to more clustered graphs and therefore better recommendation performance) when used in such contexts. It will of course be of significant interest to mitigate such adverse outcomes by well-designed interventions or by considering multiple similarity graphs that capture various dimensions of similarity. This is a compelling avenue for future work.

# B    Parameter estimation

At any time $T$, GRUB, along with the graph-side information, uses data gathered to estimate the mean $\hat{\boldsymbol{\mu}}_T$ in order to decide the sampling and elimination protocols. The following lemma gives the estimation routine used for GRUB.

**Lemma B.1.** *The closed form expression of $\hat{\boldsymbol{\mu}}_T$ is given by,*

$$\hat{\boldsymbol{\mu}}_T = \left( \sum_{t=1}^{T} \mathbf{e}_{\pi_t} \mathbf{e}_{\pi_t}^T + \rho L_G \right)^{-1} \left( \sum_{t=1}^{T} \mathbf{e}_{\pi_t} r_t^{\pi_t} \right) \tag{11}$$

*Proof.* Using the reward data $\{r_{t,\pi_t}\}_{t=1}^{T}$ gathered up-to time $T$ and the sampling policy $\boldsymbol{\pi}_T$, the mean vector estimate $\hat{\boldsymbol{\mu}}_T$ is computed by solving the following laplacian-regularized least-square optimization schedule:

$$\hat{\boldsymbol{\mu}}_T = \arg\min_{\boldsymbol{\mu} \in \mathbb{R}^n} \sum_{t=1}^{T} (\mu_{\pi_t} - r_{t,\pi_t})^2 + \rho \langle \boldsymbol{\mu}, L_G \boldsymbol{\mu} \rangle \tag{12}$$

where $\rho > 0$ is a tunable penalty parameter. The above optimization problem can be equivalently written in the following quadratic form:

$$\hat{\boldsymbol{\mu}}_T = \arg\min_{\boldsymbol{\mu} \in \mathbb{R}^n} \left( \langle \boldsymbol{\mu}, V(\boldsymbol{\pi}_T, G) \boldsymbol{\mu} \rangle - 2 \left\langle \boldsymbol{\mu}, \left( \sum_{t=1}^{T} \mathbf{e}_{\pi_t} r_{t,\pi_t} \right) \right\rangle + \sum_{t=1}^{T} r_{t,\pi_t}^2 \right)$$

where $V(\boldsymbol{\pi}_T, G)$ denotes,

$$V(\boldsymbol{\pi}_T, G) = \sum_{t=1}^{T} \mathbf{e}_{\pi_t} \mathbf{e}_{\pi_t}^T + \rho L_G \tag{13}$$

In order to obtain $\hat{\boldsymbol{\mu}}_T$, we compute vanishing point of the gradient as follows,

$$\left( \langle \boldsymbol{\mu}, V(\boldsymbol{\pi}_T, G) \boldsymbol{\mu} \rangle - 2 \left\langle \boldsymbol{\mu}, \left( \sum_{t=1}^{T} \mathbf{e}_{\pi_t} r_{t,\pi_t} \right) \right\rangle + \sum_{t=1}^{T} r_{t,\pi_t}^2 \right) |_{\boldsymbol{\mu} = \hat{\boldsymbol{\mu}}_T} = 0$$

$$\Rightarrow \quad \hat{\boldsymbol{\mu}}_T = V(\boldsymbol{\pi}_T, G)^{-1} \left( \sum_{t=1}^{T} \mathbf{e}_{\pi_t} r_t^{\pi_t} \right) \tag{14}$$

$\square$

The sampling policy in GRUB uses the mean estimates and their high probability confidence bounds to eliminate suboptimal arm. In the following lemma we compute the high probability confidence bounds on the estimates of the mean and introduces the idea of effective samples of each arm given the graph side information.

**Lemma B.2.** *For any $T > k(G)$ and $i \in [n]$, the following holds with probability no less than $1 - \frac{\delta}{w_i(\boldsymbol{\pi}_T)}$:*

$$|\hat{\mu}_T^i - \mu_i| \leq \sqrt{\frac{1}{t_{eff,i}}} \left( 2\sigma \sqrt{14 \log \left( \frac{2w_i(\boldsymbol{\pi}_T)}{\delta} \right)} + \rho \|\boldsymbol{\mu}\|_G \right) \tag{15}$$

*where $w_i(\boldsymbol{\pi}_T) = a_0 n t_{\text{eff},i}^2$ for some constant $a_0 > 0$, $\hat{\mu}_T^i$ is the $i$-th coordinate of the estimate from B.1 and,*

$$t_{\text{eff},i} = \frac{1}{\left[\left(\sum_{t=1}^T \mathbf{e}_{\pi_t}\mathbf{e}_{\pi_t}^\top + \rho L_G\right)^{-1}\right]_{ii}}$$

*Proof.* Let the sequence of bounded variance noise and data gathered up-to time $T$ be denoted by $\{\eta_t, r_{\pi_t,t}\}_{t=1}^T$. Let $S_T = \sum_{t=1}^T \eta_t \mathbf{e}_{\pi_t}$ and $N_T = \sum_{t=1}^T \mathbf{e}_{\pi_t}\mathbf{e}_{\pi_t}^T$. Using the closed form expression of $\hat{\boldsymbol{\mu}}_T$ from eq. B.1, the difference between the estimate and true value $\hat{\mu}_T^i - \mu_i$ can be obtained as follows:

$$\hat{\mu}_T^i - \mu_i = \langle \mathbf{e}_i, \hat{\boldsymbol{\mu}}_T - \boldsymbol{\mu}\rangle = \langle \mathbf{e}_i, V_T^{-1}S_T - \rho V_T^{-1}L_G\boldsymbol{\mu}\rangle$$

The deviation $\hat{\mu}_T^i - \mu_i$ can be upper-bounded as follows:

$$|\langle \mathbf{e}_i, \hat{\boldsymbol{\mu}}_T - \boldsymbol{\mu}\rangle| \leq |\langle \mathbf{e}_i, V_T^{-1}S_T\rangle| + |\langle \mathbf{e}_i, \rho V_T^{-1}L_G\boldsymbol{\mu}\rangle|$$

Further, in order to obtain the variance of the estimate $\hat{\boldsymbol{\mu}}_T$, we bound the deviation $|\mu_T^i - \mu_i|$ by separately bounding $|\langle \mathbf{e}_i, V_T^{-1}S_T\rangle|$ and $|\langle \mathbf{e}_i \rho V_T^{-1}L_G\boldsymbol{\mu}\rangle|$.

With regards to the first term $\langle \mathbf{e}_i, V_T^{-1}S_T\rangle$, note that

$$\langle \mathbf{e}_i, V_T^{-1}S_T\rangle = \left\langle \mathbf{e}_i, V_T^{-1}\left(\sum_{t=1}^T \mathbf{e}_{\pi_t}\eta_t\right)\right\rangle$$

$$= \sum_{t=1}^T \left\langle \mathbf{e}_i, V_T^{-1}\mathbf{e}_{\pi_t}\right\rangle \eta_t$$

Using a variant of Azuma's inequality [47, 51], for any $\kappa > 0$ the following inequality holds,

$$\mathbb{P}\left(|\langle \mathbf{e}_i, V_T^{-1}S_T\rangle|^2 \leq \kappa^2\right) \geq 1 - 2\exp\left\{-\frac{\kappa^2}{56\sigma^2 \sum_{t=1}^T \left(\langle \mathbf{e}_i, V_T^{-1}\mathbf{e}_{\pi_t}\rangle\right)^2}\right\} \tag{16}$$

Using the fact that $V_T \succ \left(\sum_{t=1}^T \mathbf{e}_{\pi_t}\mathbf{e}_{\pi_t}^T\right)$, we can further simplify the above bound using the following computation,

$$\sum_{t=1}^T \left(\langle \mathbf{e}_i, V_T^{-1}\mathbf{e}_{\pi_t}\rangle\right)^2 = \left\langle V_T^{-1}\mathbf{e}_i, \left(\sum_{t=1}^T \mathbf{e}_{\pi_t}\mathbf{e}_{\pi_t}^T\right)V_T^{-1}\mathbf{e}_i\right\rangle$$

$$\leq \langle \mathbf{e}_i, V_T^{-1}\mathbf{e}_i\rangle = [V_T^{-1}]_{ii} \tag{17}$$

Substituting $\delta' = 2\exp\left\{-\frac{\kappa^2}{56\sigma^2 \sum_{t=1}^T \left(\langle \mathbf{e}_i, V_T^{-1}\mathbf{e}_{\pi_t}\rangle\right)^2}\right\}$, we can finally conclude that given the historical data $\mathcal{F}_{T-1}$ till time $T-1$, following is true with probability $1 - \delta'$,

$$|\langle \mathbf{e}_i, V_T^{-1}S_T\rangle|^2 \leq 56\sigma^2 [V_T^{-1}]_{ii}\log\left(\frac{2}{\delta'}\right) \tag{18}$$

Second term $\langle \mathbf{e}_i, \rho V_T^{-1}L_G\boldsymbol{\mu}\rangle$ can be upperbounded using cauchy-schwartz inequality,

$$|\langle \mathbf{e}_i, \rho V_T^{-1}L_G\boldsymbol{\mu}\rangle| = \rho\langle \mathbf{e}_i, L_G\boldsymbol{\mu}\rangle_{V_T^{-1}}$$

$$\leq \rho\sqrt{\langle \mathbf{e}_i, V_T^{-1}\mathbf{e}_i\rangle}\sqrt{\langle L_G\boldsymbol{\mu}, V_T^{-1}L_G\boldsymbol{\mu}\rangle}$$

$$\leq \rho\sqrt{[V_T^{-1}]_{ii}}\|\boldsymbol{\mu}\|_G \tag{19}$$

Combining the upperbound (19), (18) and substituting $\delta' = \frac{\delta}{w(\boldsymbol{\pi}_T)}$ we get Lemma 3.2. Hence proved. $\qquad\square$

## C   Influence Factor

A key component in our characterization of the performance of GRUB is the *influence factor* for each arm; recall that for a given graph $D$, $C_i(D)$ denotes the connected component that contains $i$. The influence factor for each arm is defined as,

**Definition C.1.** Let $D$ be a graph on the vertex set $[n]$. For each $j \in [n]$, define **influence factor** $\Im(j, D)$ as:

$$\Im(j, D) = \begin{cases} \min_{i \in C_j(D), i \neq j} \{r_D(i, j)^{-1}\} & \text{if } |C_j(D)| > 1 \\ 0 & \text{otherwise} \end{cases} \tag{20}$$

where, $r_D(i, j)$ is the resistance distance between arm $i$ and $j$ on graph $D$ as in Definition 4.1.

Note that we refer the resistance distance without the parameter $\delta$, as the value of resistance distance is independent of the value of $\delta$. This happens due to the cancellation of $\delta$ factor in $R_{ii} + R_{jj} - R_{ji} - R_{ij}$. The influence factor can also be thought of as the minimum influence any arm $i$ in the connected component of arm $j$ has over the arm $j$

## D   Effective Samples

**Theorem D.1.** *Let $\pi_T$ indicate the sampling policy until time $T$. Let $G$ be the given graph, $\Im(., G)$ indicates the minimum influence factor for arms. Then effective samples can be lower bounded by,*

$$t_{eff,i} \geq t_i + \frac{1}{2} \lfloor \min\{\rho\Im(i, G), \sum_{j \in C(i)} t_j\} \rfloor \tag{21}$$

*where $t_i$ indicates the no. of samples of arm $i$ and $\lfloor \cdot \rfloor$ indicates the floor.*

*Proof.* Using Lemma I.5, we have the following bound on $[V_T^{-1}]_{ii}$,

$$[V(\pi_T, G)^{-1}]_{ii} \leq \max\left\{ \frac{1}{t_i + \frac{\rho\Im(i,G)}{2}}, \frac{1}{t_i + \frac{t_C - t_i}{2}} \right\} \tag{22}$$

where $T$ is the total number of samples and $t_C$ is all the samples from the connected component $C(i)$ apart from arm $i$. Thus rewriting the equation for $t_{\text{eff},i}$, we get,

$$t_{\text{eff},i} \geq t_i + \frac{1}{2} \min\{\rho\Im(i, G), \sum_{j \in C(i)} t_j\} \tag{23}$$

Hence proved.  □

## E   GRUB Sample complexity

In order to compute the sample complexity for GRUB, we classify the arms into two categories: competitive and non-competitive. The split of arms into these two categories is not required for the algorithm, but provides tighter complexity bounds as will be observed in this appendix. The division of the arms is contingent on its suboptimality and the structure of the provided graph side information. A modified version of the Definition (4.3) of competitive set and non-competitive set is as follows:

**Definition E.1.** Fix $\mu \in \mathbb{R}^n$, graph $D$, regularization parameter $\rho$, confidence parameter $\delta$, and smoothness parameter $\epsilon$ and noise variance $\sigma$. We define $\mathcal{H}$ to be the set of competitive arms and $\mathcal{N}$ to be the set of non-competitive arms as follows:

$$\mathcal{H}(D, \mu, \delta, \rho, \epsilon) = \left\{ j \in [n] \Big| \Delta_i \leq 2\sqrt{\frac{2}{\rho\Im(i)}} \left( 2\sigma\sqrt{14 \log\left(\frac{2a_0 n\rho^2\Im(i)^2}{\delta}\right)} + \rho\epsilon \right) \right\},$$

$$\mathcal{N}(D, \mu, \delta, \rho, \epsilon) \triangleq [n] \setminus \mathcal{H}(D, \mu, \delta, \rho, \epsilon)$$

When the context is clear, we will use suppress the dependence on the parameters in Definition E.1.

Further, we derive an expression for the worst-case sample complexity by analysing the number of samples required to eliminate arms with different difficulty levels, i.e. arms in competitive set and non-competitive set. We first derive the sample complexity results for the case when graph $G$ is connected and then extend it to disconnected graphs.

**Lemma E.2.** *Consider $n$-armed bandit problem with mean vector $\boldsymbol{\mu} \in \mathbb{R}^n$. Let $G$ be a given connected similarity graph on the vertex set $[n]$, and further suppose that $\boldsymbol{\mu}$ is $\epsilon$-smooth. Define*

$$T_{sufficient} \triangleq \sum_{i \in \mathcal{H}} \frac{1}{\Delta_i^2} \left[ c_1 \log \frac{c_2}{\delta \Delta_i} + \frac{\rho \epsilon}{2} \right] + \max_{i \in \mathcal{N}} \left\{ \frac{2}{\Delta_i^2} \left[ c_1 \log \frac{c_2}{\delta \Delta_i} + \frac{\rho \epsilon}{2} \right] \right\} \tag{24}$$

*Then, with probability at least $1 - \delta$, GRUB: (a) terminates in no more than $T_{sufficient}$ rounds, and (b) returns the best arm $a^* = \arg \max_i \mu_i$.*

*Proof.* With out loss of generality, assume that $a^* = 1$. Let $\{t_i\}_{i=1}^n$ denote the number of plays of each arm upto time $T$. By Lemma 3.2, we can state that,

$$\mathbb{P}\left( |\hat{\mu}_T^i - \mu_i| \geq \gamma_i(\boldsymbol{\pi}_T) \right) \leq \frac{2\delta}{a_0 n t_{\text{eff},i}^2} \tag{25}$$

where, $\gamma_i(\boldsymbol{\pi}_T) = \beta_i(\boldsymbol{\pi}_T) \sqrt{t_{\text{eff},i}^{-1}}$ and $\beta_i(\boldsymbol{\pi}_T) = \left( 2\sigma \sqrt{14 \log \left( \frac{2a_0 n t_{\text{eff},i}^2}{\delta} \right)} + \rho \|\boldsymbol{\mu}\|_G \right)$.

As is reflected in the elimination policy (4), at any time $t$, arm 1 can be mistakenly eliminated in GRUB only if $\hat{\mu}_t^i > \hat{\mu}_t^1 + \gamma_i(\boldsymbol{\pi}_t) + \gamma_1(\boldsymbol{\pi}_t)$. Let $T_s$ be the stopping time of GRUB, then the total failure probability for GRUB can be upper-bounded as,

$$\mathbb{P}(\text{Failure}) \leq \sum_{t=2}^{T_s} \sum_{i=2}^{n} \mathbb{P}\left( \hat{\mu}_t^i \geq \hat{\mu}_t^1 + \gamma_i(\boldsymbol{\pi}_t) + \gamma_1(\boldsymbol{\pi}_t) \right)$$

Note that $\mathbb{P}\left( \hat{\mu}_t^i \geq \hat{\mu}_t^1 + \gamma_i(\boldsymbol{\pi}_t) + \gamma_1(\boldsymbol{\pi}_t) \right) \leq \left[ \mathbb{P}\left( \hat{\mu}_t^i \geq \mu^i + \gamma_i(\boldsymbol{\pi}_t) \right) + \mathbb{P}\left( \hat{\mu}_t^1 \leq \mu^1 - \gamma_1(\boldsymbol{\pi}_t) \right) \right]$, provided that $\gamma_i(\boldsymbol{\pi}_t), \gamma_1(\boldsymbol{\pi}_t) \leq \frac{\Delta_i}{2}$. Hence the failure probability can be upperbounded as,

$$\mathbb{P}(\text{Failure}) \leq \sum_{i=2}^{n} \sum_{t=2}^{T_s} \left[ \mathbb{P}\left( \hat{\mu}_t^i \geq \mu^i + \gamma_i(\boldsymbol{\pi}_t) \right) + \mathbb{P}\left( \hat{\mu}_t^1 \leq \mu^1 - \gamma_1(\boldsymbol{\pi}_t) \right) \right] \tag{26}$$

conditioned on $\gamma_i(\boldsymbol{\pi}_T), \gamma_1(\boldsymbol{\pi}_T) \leq \frac{\Delta_i}{2}$.

Let $a_0 \geq 4 \sum_{t=1}^{\infty} t_{\text{eff},i}^{-2}$, then from Lemma 3.2,

$$\begin{aligned} \mathbb{P}(\text{Failure}) &\leq \sum_{i=2}^{n} \sum_{t=2}^{T_s} \frac{2\delta}{a_0 n t_{\text{eff},i}^2} \\ &\leq \delta \end{aligned} \tag{27}$$

The finiteness of the infinite sum of $t_{\text{eff},i}^{-2}$ can be found in Lemma I.13.

Thus, in order to keep $\mathbb{P}(\text{Failure}) \leq \delta$, it is sufficient if, at the time of elimination of arm $i$, we have enough samples to ensure,

$$\gamma_i(\boldsymbol{\pi}_T) \leq \frac{\Delta_i}{2}$$

$$\sqrt{\frac{1}{t_{\text{eff},i}}} \left( 2\sigma \sqrt{14 \log \left( \frac{2a_0 n t_{\text{eff},i}^2}{\delta} \right)} + \rho \epsilon \right) \leq \frac{\Delta_i}{2} \tag{28}$$

In the absence of graph information, equation (28) devolves to the same sufficiency condition for number of samples required for suboptimal arm elimination as [13], upto constant factor. Rewriting the above equation,

$$\frac{\log (a_i)}{a_i} \leq \sqrt{\frac{\delta}{d_1}} \frac{\Delta_i^2}{d_0} \tag{29}$$

where $d_0 = 64 \times 14\sigma^2, d_1 = 2na_0 e^{\frac{\rho^2\epsilon^2}{4\times14\sigma^2}}$ and $a_i = \sqrt{\frac{d_1}{\delta}}t_{\text{eff},i}$. The following bound on $a_i$ is sufficient to satisfy eq. (29),

$$a_i \geq 2\sqrt{\frac{d_1}{\delta}\frac{d_0}{\Delta_i^2}}\log\left(\sqrt{\frac{d_1}{\delta}\frac{d_0}{\Delta_i^2}}\right)$$

Resubstituting $t_{\text{eff},i}$, we obtain the sufficient number of plays required to eliminate arm $i$ as,

$$t_{\text{eff},i} \geq \frac{c_1}{\Delta_i^2}\left[\log\left(\frac{c_2}{\delta^{\frac{1}{2}}\Delta_i^2}\right) + c_3\right] \tag{30}$$

where $c_1 = 2 \times 64 \times 14\sigma^2$, $c_2 = 64 \times 14\sigma^2\sqrt{2na_0}$ and $c_3 = \frac{\rho^2\epsilon^2}{8\times14\sigma^2}$. In the further text we are suppressing the powers of $\delta, \Delta_i$ within the log factor as it adds only a constant multiple to the lower bound.

The further part of the proof we use the following bound on $t_{\text{eff},.}$ from Theorem D.1 as follows:

$$t_{\text{eff},i} \geq t_i + \frac{1}{2}\min\left\{\rho\mathfrak{I}(i), T - t_i\right\} \quad \forall i \in [n] \tag{31}$$

Hence a sufficiency condition for the GRUB to produce the best-arm with probability $1 - \delta$ is given when both the following conditions are satisfied,

$$t_i + \frac{\rho\mathfrak{I}(i)}{2} \geq \frac{1}{\Delta_i^2}\left[c_1\log\left(\frac{c_2}{\delta\Delta_i}\right) + \frac{\rho\epsilon}{2}\right] \tag{32}$$

and,

$$T + t_i \geq T \geq \frac{2}{\Delta_i^2}\left[c_1\log\left(\frac{c_2}{\delta\Delta_i}\right) + \frac{\rho\epsilon}{2}\right] \tag{33}$$

From the Definition E.1 of competitive arms $\mathcal{H}$ and non-competitive arms $\mathcal{N}$, we have,

$$\mathcal{H} = \left\{j \in [n] | \Delta_i \leq 2\sqrt{\frac{2}{\rho\mathfrak{I}(i)}}\left(2\sigma\sqrt{14\log\left(\frac{2a_0n\rho^2\mathfrak{I}(i)^2}{\delta}\right)} + \rho\epsilon\right)\right\} \tag{34}$$

After the first $\max_{i\in\mathcal{N}}\left\{\frac{2}{\Delta_i^2}\left[c_1\log\frac{c_2}{\delta\Delta_i} + \frac{\rho\epsilon}{2}\right]\right\}$ samples, all arms in $\mathcal{N}$ are eliminated. Further, let $k_1$ be the index of the first arm to be eliminated (in $\mathcal{H}$) and $t_{k_1}^*$ be the number of samples of arm $k_1$ before getting eliminated then the total number of additional time steps played until the arm $k_1$ is eliminated is at most $|\mathcal{H}|t_{k_1}^*$. Let $k_2$ be the index of the next arm in $\mathcal{H}$ to be eliminated. The number of additional plays until the next arm is eliminated is given by $(|\mathcal{H}| - 1)[t_{k_2}^* - t_{k_1}^*]$ and so on.

Summing up all the samples required to converge to the optimal arm is given by, (let $t_{k_0}^* = 0$)

$$\sum_{h=1}^{|\mathcal{H}|}(|\mathcal{H}| - h))[t_{k_h}^* - t_{k_{h-1}}^*] = \sum_{h=1}^{|\mathcal{H}|-1} t_{k_h}^* = \sum_{i\in\mathcal{H}/1} t_i^* \tag{35}$$

Hence the final sample complexity can be computed as follows:

- Number of plays required for arms in $\mathcal{H}$ :

$$\sum_{i\in\mathcal{H}/1} t_i^* \geq \sum_{i\in\mathcal{H}/1} \frac{1}{\Delta_i^2}\left[c_1\log\frac{c_2}{\delta\Delta_i} + \frac{\rho\epsilon}{2}\right] \tag{36}$$

- Number of plays required for all the arms in $\mathcal{N} := [n]/\mathcal{H}$ to be eliminated:

$$T \geq \max_{i\in\mathcal{N}}\left\{\frac{2}{\Delta_i^2}\left[c_1\log\frac{c_2}{\delta\Delta_i} + \frac{\rho\epsilon}{2}\right]\right\} \tag{37}$$

Hence the final sample complexity can be given by,

$$T_{\text{sufficient}} \triangleq \max_{i \in \mathcal{N}} \left\{ \frac{2}{\Delta_i^2} \left[ c_1 \log \frac{c_2}{\delta \Delta_i} + \frac{\rho \epsilon}{2} \right] \right\} + \sum_{i \in \mathcal{H}/1} \frac{1}{\Delta_i^2} \left[ c_1 \log \frac{c_2}{\delta \Delta_i} + \frac{\rho \epsilon}{2} \right] \tag{38}$$

Hence proved. □

We extend Lemma E.2 to the case when graph $G$ has disconnected clusters.

**Note:** The following theorem stated in the main paper has a typographical error in the equation for $T_{\text{sufficient}}$ in place of $\arg \min$ it is supposed to be $\min$.

**Theorem E.3.** *Consider $n$-armed bandit problem with mean vector $\boldsymbol{\mu} \in \mathbb{R}^n$. Let $\mathcal{G}$ be the set of subgraphs of given similarity graph $G$ on the vertex set $[n]$, and further suppose that $\boldsymbol{\mu}$ is $\epsilon$-smooth. Define*

$$T_{\text{sufficient}} \triangleq \min_{D \in \mathcal{G}} \sum_{C \in \mathcal{C}_D} \left[ \sum_{i \in C \cap \mathcal{H}_D} \frac{1}{\Delta_i^2} \left[ c_1 \log \frac{c_2}{\delta \Delta_i} + \frac{\rho \epsilon}{2} \right] + \max_{i \in C \cap \mathcal{N}_D} \left\{ \frac{2}{\Delta_i^2} \left[ c_1 \log \frac{c_2}{\delta \Delta_i} + \frac{\rho \epsilon}{2} \right] \right\} \right] \tag{39}$$

*where $\Delta_i = \mu^* - \mu_i$ for all suboptimal arms, $\mathcal{H}_D$ and $\mathcal{N}_D$ are as in Definition E.1, $\mathcal{C}_D$ is the set of connected components of a subgraph $D \in \mathcal{G}$ and $c_1, c_2$ are constants independent of system parameters. Then, with probability at least $1 - \delta$, GRUB: (a) terminates in no more than $T_{\text{sufficient}}$ rounds, and (b) returns the best arm $a^* = \arg \max_i \mu_i$.*

*Proof.* Let $\mathcal{C}_G$ denote the connected components of graph $G$. From Lemma E.2, the number of samples for each connected component $C \in \mathcal{C}_G$ can be given as,

$$T_{\text{sufficient}} = \left[ \sum_{i \in C \cap \mathcal{H}} \frac{1}{\Delta_i^2} \left[ c_1 \log \frac{c_2}{\delta \Delta_i} + \frac{\rho \epsilon}{2} \right] + \max_{i \in C \cap \mathcal{N}} \left\{ \frac{2}{\Delta_i^2} \left[ c_1 \log \frac{c_2}{\delta \Delta_i} + \frac{\rho \epsilon}{2} \right] \right\} \right] \tag{40}$$

We can obtain the sample complexity for obtaining the best arm by summing it over all the components $C \in \mathcal{C}$, gives us the sample complexity for GRUB while considering graph $G$.

$$T_{\text{sufficient}} = \sum_{C \in \mathcal{C}_G} \left[ \sum_{i \in C \cap \mathcal{H}} \frac{1}{\Delta_i^2} \left[ c_1 \log \frac{c_2}{\delta \Delta_i} + \frac{\rho \epsilon}{2} \right] + \max_{i \in C \cap \mathcal{N}} \left\{ \frac{2}{\Delta_i^2} \left[ c_1 \log \frac{c_2}{\delta \Delta_i} + \frac{\rho \epsilon}{2} \right] \right\} \right] \tag{41}$$

Any subgraph $D$ of graph $G$ satisfies,

$$\langle \boldsymbol{\mu}, L_G \boldsymbol{\mu} \rangle \le \epsilon \Rightarrow \langle \boldsymbol{\mu}, L_D \boldsymbol{\mu} \rangle \le \epsilon \tag{42}$$

As seen in Definition E.1, the influence factor is instrumental in deciding the competitive and non-competitive sets, which further dictates the sample complexity bounds. Further, notice from Lemma I.8 that the influence factor $\mathfrak{I}(i, D)$ is not monotonic when considering subgraph $D$ of graph $G$. Hence considering a subgraph of $G$ could potentially increase the number of non-competitive arms and provide us with a tighter bound on the performance for GRUB.

Hence $T_{\text{sufficient}}$ in (40) can be made tighter by considering the minimum value over the entire set of subgraphs $\mathcal{G}$. □

We next derive sample complexity upper bounds for GRUB in certain illuminating special cases.

**Corollary E.4** (Isolated clusters). *Consider the setup as in Theorem 4.4 with the further restriction that $G$ consists of a subgraph $F$ such that optimal node is isolated and arms $[2, \ldots, n]$ are split in $k$ clusters and $\Delta_i \ge 2\sqrt{\frac{2}{\rho \mathfrak{I}(i,F)}} \left( 2\sigma \sqrt{14 \log \left( \frac{2a_0 n \rho^2 \mathfrak{I}(i,F)^2}{\delta} \right)} + \rho \epsilon \right), \forall i \in [2, \ldots, n]$. Define*

$$T_{\text{sufficient}} \triangleq \sum_{C \in \mathcal{C}_F/1} \max_{j \in C} \frac{2}{\Delta_j^2} \left[ c_1 \log \left( \frac{c_2}{\delta \Delta_i} \right) + \frac{\rho \epsilon}{2} \right] \tag{43}$$

*Then, with probability at least $1 - \delta$, GRUB: (a) terminates in no more than $T_{\text{sufficient}}$ rounds, and (b) returns the best arm $a^* = \arg \max_i \mu_i$.*

---

**Algorithm 1** GRUB

**Input:** Regularization parameter $\rho$, Smoothness parameter $\epsilon$, Error bound $\delta$, Total arms $n$, Laplacian $L_G$, Sub-gaussianity parameter $\sigma$

$t \leftarrow 0$
$A = \{1, 2, \ldots, n\}$
$t = 0$
$V_0 \leftarrow \rho L_G$
$\mathcal{C}(G) \leftarrow \texttt{Cluster-Identification}(L_G)$
**for** $C \in \mathcal{C}(G)$ **do**
$\quad t \leftarrow t + 1$
$\quad$ Pick random arm $k \in C$ to observe reward $r_{t,k}$
$\quad V_t \leftarrow V_{t-1} + \mathbf{e}_k \mathbf{e}_k^T$, and $\mathbf{x}_t \leftarrow \mathbf{x}_{t-1} + r_{t,k} \mathbf{e}_k$
**end for**
**while** $|A| > 1$ **do**
$\quad t \leftarrow t + 1$
$\quad$ **for** $i \in A$ **do**
$\quad\quad t_{\text{eff},i} \leftarrow ([V_t^{-1}]_{ii})^{-1}$
$\quad\quad \beta_i(t) \leftarrow 2\sigma \sqrt{14 \log\left(\frac{2n t_{\text{eff},i}^2}{\delta}\right)} + \rho\epsilon$
$\quad$ **end for**
$\quad k \leftarrow \texttt{Sampling-Policy}(t, V_t, A, \mathcal{C}(G))$
$\quad$ Sample arm $k$ to observe reward $r_{t,k}$
$\quad V_t \leftarrow V_{t-1} + \mathbf{e}_k \mathbf{e}_k^T$
$\quad \mathbf{x}_t \leftarrow \mathbf{x}_{t-1} + r_{t,k} \mathbf{e}_k$
$\quad \hat{\boldsymbol{\mu}}_t \leftarrow V_t^{-1} \mathbf{x}_t$
$\quad a_{\max} \leftarrow \arg\max_{i \in A} \left[ \hat{\mu}_t^i - \beta(t_i) \sqrt{t_{\text{eff},i}^{-1}} \right]$
$\quad A \leftarrow \left\{ \mathbf{a} \in A \mid \hat{\mu}_t^{a_{\max}} - \hat{\mu}_t^a \leq \beta_a(t) \sqrt{t_{\text{eff},a}^{-1}} \right.$
$\quad\quad\quad \left. + \beta_{a_{\max}}(t) \sqrt{t_{\text{eff},a_{\max}}^{-1}} \right\}$
**end while**
**return** A

---

Corollary E.4 shows that in scenarios where the arms are well clustered, the sample complexity of GRUB can scale with the number of clusters, a quantity that is typically significantly smaller than the total number of nodes in the graph.

**Corollary E.5** (Star graph). *Consider the setup as in Theorem 4.4 with the further restriction that $G$ consists of a star subgraph with the central node as the optimal arm and $\Delta_i \leq$*

$2\sqrt{\frac{2}{\rho \mathfrak{I}(i,F)}} \left( 2\sigma \sqrt{14 \log\left(\frac{2a_0 n \rho^2 \mathfrak{I}(i,F)^2}{\delta}\right)} + \rho\epsilon \right), \forall i \in [2, \ldots, n]$. *Define*

$$T_{\text{sufficient}} \triangleq \sum_{i=2}^{n} \frac{1}{\Delta_i^2} \left[ c_1 \log\left(\frac{c_2}{\delta \Delta_i}\right) + \frac{\rho\epsilon}{2} \right] \tag{44}$$

*Then, with probability at least $1 - \delta$, GRUB: (a) terminates in no more than $T_{\text{sufficient}}$ rounds, and (b) returns the best arm $a^* = \arg\max_i \mu_i$.*

In Corollary E.5, $T_{\text{sufficient}}$ is the same sample complexity as vanilla best arm identification, upto constant factors which is due to the fact that pulling one of the spoke arms does not yield much information about the other spoke arms, and this is the exact situation in the standard pure exploration setting.

# F    Lower bounds

In this section we give a lower bound on the sample complexity for any $\delta$-PAC to return the best arm for a $n$ armed bandit problem along with graph side information.

**Theorem F.1.** *Given an $n$-armed bandit model with associated mean vector $\boldsymbol{\mu} \in \mathbb{R}^n$ and similarity graph $G$ smooth on $\boldsymbol{\mu}$, i.e. $\langle \boldsymbol{\mu}, L_G \boldsymbol{\mu} \rangle \leq \epsilon$, for any $0 < \epsilon < \epsilon_0$. Let $G = ([n], E)$ be the graph with only $k$ isolated cliques and w.l.o.g let arm 1 be the optimal arm. Then define*

$$T_{necessary} = \sum_{C \in \mathcal{C}_G / C^*} \min_{j \in C} \left\{ \frac{4\sigma^2 \log 5}{(\Delta_j - \sqrt{\epsilon})^2} \right\} + \sum_{j \in C^*/1} \frac{4\sigma^2 \log 5}{\Delta_j^2} \tag{45}$$

*where $C^*$ is the clique with the optimal arm and $\epsilon_0 := \min\limits_{i \in [n]/1, j \in C(i)} \left[ \Delta_j \left[ 1 - \frac{\Delta_i}{\sqrt{\Delta_i^2 + \Delta_j^2}} \right] \right]^2$. Then any $\delta$-PAC algorithm will need at-least $T_{necessary}$ steps to terminate, provided $\delta \leq 0.1$.*

*Proof.* We prove the theorem in two steps: Firstly, we construct the sample complexity lower bound for the similarity graph with the isolated optimal arm and a clique of rest of the sub-optimal arms, followed by step 2 the sample complexity lower bound for a graph with single cluster

**Step 1:**

Consider a $n + 1$ armed bandit problem with mean vector $\boldsymbol{\mu} \in \mathbb{R}^{n+1}$ and similarity graph $M$ with an isolated optimal arm (arm 1) and $n$-clique cluster of suboptimal arms, satisfying the condition for smoothness of rewards over the graph,i.e., $\langle \boldsymbol{\mu}, L_M \boldsymbol{\mu} \rangle \leq \epsilon$. Then the following holds

$$\max_{i \neq 1} \mu_i \leq \min_{j \neq 1} \{\mu_j + \sqrt{\epsilon}\} \tag{46}$$

Assume that ordering of mean in $n$-clique of suboptimal arms is known. From [26], there exists a $\delta$-PAC algorithm, for $\delta \leq 0.1$, which can successful identify the best arm for the subproblem with just the optimal arm and arm with the maximum mean in the $n$-clique cluster, i.e. $j' = \arg\max\limits_{j \neq 1} \mu_j$

with the total number of samples given by,

$$T \geq \frac{4 \log 5 \sigma^2}{\Delta_{j'}^2} \tag{47}$$

Now consider the case where the ordering of the mean in $n$-clique is unknown. In order to remove all the suboptimal arms provided $\epsilon \leq \min_{j \neq 1} \Delta_j^2$ and (46) holds, it is suffices to be able to distinguish between the optimal arm and a hypothetical suboptimal arm with mean $\mu_j + \sqrt{\epsilon}$ where $j$ is any arm from suboptimal $n$-clique, and the minimum number of samples required by any $\delta$-PAC algorithm to successfully identify the best arm with $\delta \leq 0.1$ is given by,

$$T \geq \frac{4 \log 5 \sigma^2}{(\Delta_j - \sqrt{\epsilon})^2} \tag{48}$$

The best performance in terms of sample complexity out of all the random choice of arm from the suboptimal $n$-clique cluster is,

$$T \geq \min_{j \neq 1} \left\{ \frac{4 \log 5 \sigma^2}{(\Delta_j - \sqrt{\epsilon})^2} \right\} \tag{49}$$

Given $\epsilon_0 := \min\limits_{i \in [n]/1, j \in C(i)} \left[ \Delta_j \left[ 1 - \frac{\Delta_i}{\sqrt{\Delta_i^2 + \Delta_j^2}} \right] \right]^2$ and $\epsilon < \epsilon_0$, it can be verified that for any arm $i, j \neq 1$,

$$\min_{j \neq 1} \frac{4 \log 5 \sigma^2}{(\Delta_j - \sqrt{\epsilon})^2} < \frac{4 \log 5 \sigma^2}{\Delta_i^2} + \frac{4 \log 5 \sigma^2}{\Delta_j^2} \tag{50}$$

where the left hand side corresponds to the sample complexity lower bound of removing the suboptimal arms $i, j$ with the graph side information and the right hand side corresponds to the same without graph side information.

Hence it can be inferred that it is inefficient to remove the arms individually (disregarding the graph information).

**Step 2 :**

Consider a $n + 1$ armed bandit problem with mean vector $\boldsymbol{\mu} \in \mathbb{R}^{n+1}$ with a given similarity graph $N$ such that $\langle \boldsymbol{\mu}, L_N \boldsymbol{\mu} \rangle \leq \epsilon$. Let all the suboptimal arms be connected to the optimal arm.

Here we show by an adversarial example that it is not possible to have a lower bound on the sample complexity which scales better than,

$$T \geq \sum_{j \neq 1} \frac{4 \log 5 \sigma^2}{\Delta_j^2} \tag{51}$$

There exists a $\delta$-PAC algorithm which can determine that arms $j = 3, \ldots, n$ are suboptimal after $T \geq \sum_{j \neq 1,2} \frac{1}{\Delta_j^2}$ samples. From the smoothness of rewards on the similarity graph $N$ we know that,

$$-\sqrt{\epsilon} \leq \mu_1 - \mu_j \leq \sqrt{\epsilon} \quad \forall j \in [2, 3, \ldots, n] \tag{52}$$

This information does not help us identify or even reduce the number of samples required to identify optimal arm between arm 1 and arm 2. Thus no $\delta$-PAC algorithm, $\delta \leq 0.1$, can determine the optimal arm from arm 1 and arm 2 without an additional $\frac{4 \log 5 \sigma^2}{\Delta_2^2}$ samples for determining the best arm.

Using above two steps, we construct the proof for lower bound as follows:

Now consider the graph side information as defined in the theorem, and let $\mathcal{C}_G$ denote the set of connected components of graph $G$ and $C^* \in \mathcal{C}_G$ be the component containing the optimal arm. Finding the best arm in this setup requires elimination of the suboptimal arms with in the connected component containing optimal arm $j \in C^*$ and elimination of the other connected components with suboptimal arms $j \in \mathcal{C}_G/C^*$. Hence, the sample complexity lower bounds [26, 27] for any $\delta$-PAC algorithm with $\delta \leq 0.1$ to eliminate these arms using the tools developed in step 1 and step 2, is given by

$$T \geq \sum_{j \in C^*/1} \frac{4 \sigma^2 \log 5}{\Delta_j^2} + \sum_{C \in \mathcal{C}_G/C^*} \min_{j \in C} \left\{ \frac{4 \sigma^2 \log 5}{(\Delta_j - \sqrt{\epsilon})^2} \right\} \tag{53}$$

$\square$

# G $\quad \zeta$-GRUB Sample complexity proof

**Definition G.1.** Fix $\boldsymbol{\mu} \in \mathbb{R}^n$, graph $D$, confidence parameter $\delta$, noise variance $\sigma$, and relaxation parameter $\zeta$. We define $\mathcal{H}$ to be the set of competitive arms and $\mathcal{N}$ to be the set of non-competitive arms for $\zeta$-GRUB as follows:

$$\mathcal{H}(D, \boldsymbol{\mu}, \delta, \zeta) = \left\{ j \in [n] \Big| \Delta_i^\zeta \leq 2 \sqrt{\frac{2}{\rho \mathfrak{J}(i)}} \left( 2\sigma \sqrt{14 \log \left( \frac{2 a_0 n \rho^2 \mathfrak{J}(i)^2}{\delta} \right)} + \rho \epsilon \right) \right\},$$

$$\mathcal{N}(D, \boldsymbol{\mu}, \delta, \zeta) \triangleq [n] \setminus \mathcal{H}(D, \boldsymbol{\mu}, \delta, \zeta)$$

where $\Delta_i^\zeta \triangleq \max\{\Delta_i, \zeta\}$.

**Lemma G.2.** *Consider $n$-armed bandit problem with mean vector $\boldsymbol{\mu} \in \mathbb{R}^n$. Let $G$ be a given connected similarity graph on the vertex set $[n]$, and further suppose that $\boldsymbol{\mu}$ is $\epsilon$-smooth. Define*

$$T_{\text{sufficient}} \triangleq \sum_{i \in \mathcal{H}} \frac{1}{(\Delta_i^\zeta)^2} \left[ c_1 \log \frac{c_2}{\delta \Delta_i^\zeta} + \frac{\rho \epsilon}{2} \right] + \max_{i \in \mathcal{N}} \left\{ \frac{2}{(\Delta_i^\zeta)^2} \left[ c_1 \log \frac{c_2}{\delta \Delta_i^\zeta} + \frac{\rho \epsilon}{2} \right] \right\} \tag{54}$$

*where $\Delta_i^\zeta \triangleq \max\{\Delta_i, \zeta\}$. Then, with probability at least $1 - \delta$, GRUB: (a) terminates in no more than $T_{\text{sufficient}}$ rounds, and (b) returns a $\zeta$-best arm*

*Proof.* With out loss of generality, assume that $a^* = 1$. Let $\{t_i\}_{i=1}^n$ denote the number of plays of each arm upto time $T$. By Lemma 3.2, we can state that,

$$\mathbb{P}\left( |\hat{\mu}_T^i - \mu_i| \geq \gamma_i(\boldsymbol{\pi}_T) \right) \leq \frac{2\delta}{a_0 n t_{\text{eff}, i}^2} \tag{55}$$

where, $\gamma_i(\boldsymbol{\pi}_T) = \beta_i(\boldsymbol{\pi}_T)\sqrt{t_{\text{eff},i}^{-1}}$ and $\beta_i(\boldsymbol{\pi}_T) = \left(2\sigma\sqrt{14\log\left(\frac{2a_0 n t_{\text{eff},i}^2}{\delta}\right)} + \rho\|\boldsymbol{\mu}\|_G\right)$.

As is reflected in the elimination policy (4), at any time $t$, arm 1 can be mistakenly eliminated in GRUB only if $\hat{\mu}_t^i > \hat{\mu}_t^1 + \gamma_i(\boldsymbol{\pi}_t) + \gamma_1(\boldsymbol{\pi}_t)$. Let $T_s$ be the stopping time of GRUB, then the total failure probability for GRUB can be upper-bounded as,

$$\mathbb{P}(\text{Failure}) \leq \sum_{t=2}^{T_s}\sum_{i=2}^{n} \mathbb{P}\left(\hat{\mu}_t^i \geq \hat{\mu}_t^1 + \gamma_i(\boldsymbol{\pi}_t) + \gamma_1(\boldsymbol{\pi}_t)\right)$$

Note that $\mathbb{P}\left(\hat{\mu}_t^i \geq \hat{\mu}_t^1 + \gamma_i(\boldsymbol{\pi}_t) + \gamma_1(\boldsymbol{\pi}_t)\right) \leq \left[\mathbb{P}\left(\hat{\mu}_t^i \geq \mu^i + \gamma_i(\boldsymbol{\pi}_t)\right) + \mathbb{P}\left(\hat{\mu}_t^1 \leq \mu^1 - \gamma_1(\boldsymbol{\pi}_t)\right)\right]$, provided that $\gamma_i(\boldsymbol{\pi}_t), \gamma_1(\boldsymbol{\pi}_t) \leq \frac{\Delta_i^\zeta}{2}$. Hence the failure probability can be upperbounded as,

$$\mathbb{P}(\text{Failure}) \leq \sum_{i=2}^{n}\sum_{t=2}^{T_s}\left[\mathbb{P}\left(\hat{\mu}_t^i \geq \mu^i + \gamma_i(\boldsymbol{\pi}_t)\right) + \mathbb{P}\left(\hat{\mu}_t^1 \leq \mu^1 - \gamma_1(\boldsymbol{\pi}_t)\right)\right] \tag{56}$$

conditioned on $\gamma_i(\boldsymbol{\pi}_T), \gamma_1(\boldsymbol{\pi}_T) \leq \frac{\Delta_i^\zeta}{2}$.

Let $a_0 \geq 4\sum_{t=1}^{\infty} t_{\text{eff},i}^{-2}$, then from Lemma 3.2,

$$\mathbb{P}(\text{Failure}) \leq \sum_{i=2}^{n}\sum_{t=2}^{T_s}\frac{2\delta}{a_0 n t_{\text{eff},i}^2}$$
$$\leq \delta \tag{57}$$

The finiteness of the infinite sum of $t_{\text{eff},i}^{-2}$ can be found in Lemma I.13.

Thus, in order to keep $\mathbb{P}(\text{Failure}) \leq \delta$, it is sufficient if, at the time of elimination of arm $i$, we have enough samples to ensure,

$$\gamma_i(\boldsymbol{\pi}_T) \leq \frac{\Delta_i^\zeta}{2}$$
$$\sqrt{\frac{1}{t_{\text{eff},i}}}\left(2\sigma\sqrt{14\log\left(\frac{2a_0 n t_{\text{eff},i}^2}{\delta}\right)} + \rho\epsilon\right) \leq \frac{\Delta_i^\zeta}{2} \tag{58}$$

Rewriting the above equation,

$$\frac{\log(a_i)}{a_i} \leq \sqrt{\frac{\delta}{d_1}}\frac{(\Delta_i^\zeta)^2}{d_0} \tag{59}$$

where $d_0 = 64 \times 14\sigma^2$, $d_1 = 2na_0 e^{\frac{\rho^2\epsilon^2}{4\times 14\sigma^2}}$ and $a_i = \sqrt{\frac{d_1}{\delta}}t_{\text{eff},i}$. The following bound on $a_i$ is sufficient to satisfy eq. (59),

$$a_i \geq 2\sqrt{\frac{d_1}{\delta}}\frac{d_0}{(\Delta_i^\zeta)^2}\log\left(\sqrt{\frac{d_1}{\delta}}\frac{d_0}{(\Delta_i^\zeta)^2}\right)$$

Resubstituting $t_{\text{eff},i}$, we obtain the sufficient number of plays required to eliminate arm $i$ as,

$$t_{\text{eff},i} \geq \frac{c_1}{(\Delta_i^\zeta)^2}\left[\log\left(\frac{c_2}{\delta^{\frac{1}{2}}(\Delta_i^\zeta)^2}\right) + c_3\right] \tag{60}$$

where $c_1 = 2 \times 64 \times 14\sigma^2$, $c_2 = 64 \times 14\sigma^2\sqrt{2na_0}$ and $c_3 = \frac{\rho^2\epsilon^2}{8\times 14\sigma^2}$.

The further part of the proof depends crucially on the following bound on $t_{\text{eff},i}$ for all $i \in [n]$ from Theorem D.1 as follows:

$$t_{\text{eff},i} \geq t_i + \frac{1}{2}\min\{\rho\mathfrak{I}(i), T - t_i\} \tag{61}$$

Hence a sufficiency condition for the GRUB to produce the $\zeta$-best arm with probability $1 - \delta$ is given when both the following conditions are satisfied,

$$t_i + \frac{\rho \mathfrak{I}(i)}{2} \geq \frac{1}{(\Delta_i^\zeta)^2} \left[ c_1 \log \left( \frac{c_2}{\delta \Delta_i^\zeta} \right) + \frac{\rho \epsilon}{2} \right] \tag{62}$$

and,

$$T + t_i \geq T \geq \frac{2}{(\Delta_i^\zeta)^2} \left[ c_1 \log \left( \frac{c_2}{\delta \Delta_i^\zeta} \right) + \frac{\rho \epsilon}{2} \right] \tag{63}$$

From the Definition G.1 we have the set of competitive arms $\mathcal{H}$ and non-competitive arms $\mathcal{N}$ as follows:

$$\mathcal{H} = \left\{ j \in [n] \big| \Delta_i^\zeta \leq 2 \sqrt{\frac{2}{\rho \mathfrak{I}(i)}} \left( 2\sigma \sqrt{14 \log \left( \frac{2a_0 n \rho^2 \mathfrak{I}(i)^2}{\delta} \right)} + \rho \epsilon \right) \right\} \tag{64}$$

After the first $\max_{i \in \mathcal{N}} \left\{ \frac{2}{(\Delta_i^\zeta)^2} \left[ c_1 \log \frac{c_2}{\delta \Delta_i^\zeta} + \frac{\rho \epsilon}{2} \right] \right\}$ samples, all arms in $\mathcal{N}$ are eliminated. Further, let $k_1$ be the index of the first arm to be eliminated (in $\mathcal{H}$) and $t_{k_1}^*$ be the number of samples of arm $k_1$ before getting eliminated then the total number of additional time steps played until the arm $k_1$ is eliminated is at most $|\mathcal{H}| t_{k_1}^*$. Let $k_2$ be the index of the next arm in $\mathcal{H}$ to be eliminated. The number of additional plays until the next arm is eliminated is given by $(|\mathcal{H}| - 1)[t_{k_2}^* - t_{k_1}^*]$ and so on.

Summing up all the samples required to converge to the optimal arm is given by, (let $t_{k_0}^* = 0$)

$$\sum_{h=1}^{|\mathcal{H}|} (|\mathcal{H}| - h))[t_{k_h}^* - t_{k_{h-1}}^*] = \sum_{h=1}^{|\mathcal{H}|-1} t_{k_h}^* = \sum_{i \in \mathcal{H}/1} t_i^* \tag{65}$$

Hence the final sample complexity can be computed as follows:

- Number of plays required for arms in $\mathcal{H}$ :

$$\sum_{i \in \mathcal{H}/1} t_i^* \geq \sum_{i \in \mathcal{H}/1} \frac{1}{(\Delta_i^\zeta)^2} \left[ c_1 \log \frac{c_2}{\delta \Delta_i^\zeta} + \frac{\rho \epsilon}{2} \right] \tag{66}$$

- Number of plays required for all the arms in $\mathcal{N} := [n]/\mathcal{H}$ to be eliminated:

$$T \geq \max_{i \in \mathcal{N}} \left\{ \frac{2}{(\Delta_i^\zeta)^2} \left[ c_1 \log \frac{c_2}{\delta \Delta_i^\zeta} + \frac{\rho \epsilon}{2} \right] \right\} \tag{67}$$

Hence the final sample complexity can be given by,

$$T_{\text{sufficient}} \triangleq \max_{i \in \mathcal{N}} \left\{ \frac{2}{(\Delta_i^\zeta)^2} \left[ c_1 \log \frac{c_2}{\delta \Delta_i^\zeta} + \frac{\rho \epsilon}{2} \right] \right\} + \sum_{i \in \mathcal{H}/1} \frac{1}{(\Delta_i^\zeta)^2} \left[ c_1 \log \frac{c_2}{\delta \Delta_i^\zeta} + \frac{\rho \epsilon}{2} \right] \tag{68}$$

$\square$

We extend Lemma G.2 to the case when graph $G$ has disconnected clusters.

**Note:** The following theorem stated in the main paper has a typographical error in the equation for $T_{\text{sufficient}}$ in place of $\arg \min$ it is supposed to be $\min$.

**Theorem G.3.** *Consider $n$-armed bandit problem with mean vector $\boldsymbol{\mu} \in \mathbb{R}^n$. Let $\mathcal{G}$ be the set of subgraphs given similarity graph $G$ on the vertex set $[n]$, and further suppose that $\boldsymbol{\mu}$ is $\epsilon$-smooth. Define*

$$T_{\text{sufficient}} \triangleq \min_{D \in \mathcal{G}} \sum_{C \in \mathcal{C}_D} \left[ \sum_{i \in C \cap \mathcal{H}_D} \frac{1}{(\Delta_i^\zeta)^2} \left[ c_1 \log \frac{c_2}{\delta \Delta_i^\zeta} + \frac{\rho \epsilon}{2} \right] \right.$$
$$\left. + \max_{i \in C \cap \mathcal{N}_D} \left\{ \frac{2}{(\Delta_i^\zeta)^2} \left[ c_1 \log \frac{c_2}{\delta \Delta_i^\zeta} + \frac{\rho \epsilon}{2} \right] \right\} \right] \tag{69}$$

---

**Algorithm 2** $\zeta$-GRUB

---

**Input:** Regularization parameter $\rho$, Smoothness parameter $\epsilon$, Error bound $\delta$, Total arms $n$, Laplacian $L_G$, Sub-gaussianity parameter $\sigma$

$t \leftarrow 0$

$A = \{1, 2, \ldots, n\}$

$t = 0$

$V_0 \leftarrow \rho L_G$

$\mathcal{C}(G) \leftarrow \texttt{Cluster-Identification}(L_G)$

**for** $C \in \mathcal{C}(G)$ **do**

    $t \leftarrow t + 1$

    Pick random arm $k \in C$ to observe reward $r_{t,k}$

    $V_t \leftarrow V_{t-1} + \mathbf{e}_k \mathbf{e}_k^T$, and $\mathbf{x}_t \leftarrow \mathbf{x}_{t-1} + r_{t,k} \mathbf{e}_k$

**end for**

**while** $|A| > 1$ **do**

    $t \leftarrow t + 1$

    $\beta(t) \leftarrow 2\sigma \sqrt{14 \log\left(\frac{2n(t+1)^2}{\delta}\right)} + \rho\epsilon$

    $k \leftarrow \texttt{Sampling-Policy}(t, V_t, A, \mathcal{C}(G))$

    Sample arm $k$ to observe reward $r_{t,k}$

    $V_t \leftarrow V_{t-1} + \mathbf{e}_k \mathbf{e}_k^T$

    $\mathbf{x}_t \leftarrow \mathbf{x}_{t-1} + r_{t,k} \mathbf{e}_k$

    $\hat{\boldsymbol{\mu}}_t \leftarrow V_t^{-1} \mathbf{x}_t$

    $a_{\max} \leftarrow \underset{i \in A}{\arg\max} \left[ \hat{\mu}_t^i - \beta(t_i)\sqrt{[V_t^{-1}]_{ii}} \right]$

    $A \leftarrow \left\{ \mathbf{a} \in A \mid \hat{\mu}_t^{a_{\max}} - \hat{\mu}_t^a \leq \beta(t_a)\sqrt{[V_t^{-1}]_{aa}} \right.$

             $\left. + \beta(t_{a_{\max}})\sqrt{[V_t^{-1}]_{a_{\max} a_{\max}}} \right\}$

    $A \leftarrow A / \left\{ a \in A \mid \beta(t_a)\sqrt{[V_t^{-1}]_{aa}} \leq \frac{\zeta}{2} \right\}$

**end while**

**return** $\arg\max \left\{ \mu_i \mid i \in \{a \in [n] \mid \beta(t_a)\sqrt{[V_t^{-1}]_{aa}} \leq \frac{\zeta}{2}\} \cup A \right\}$

---

*where $\Delta_i^\zeta = \max\{\Delta_i, \zeta\}$ for all suboptimal arms, $\mathcal{H}_D$ and $\mathcal{N}_D$ are as in Definition G.1, $\mathcal{C}_D$ is the set of connected components of subgraph $D \in \mathcal{G}$ and $c_1, c_2$ are constants independent of system parameters. Then, with probability at least $1 - \delta$, GRUB: (a) terminates in no more than $T_{\text{sufficient}}$ rounds, and (b) returns a $\zeta$-best arm*

*Proof.* From Lemma G.2, the sample complexity for each connected component $C \in \mathcal{C}$ can be given as,

$$T_{\text{sufficient}} = \left\lceil \sum_{i \in C \cap \mathcal{H}} \frac{1}{(\Delta_i^\zeta)^2} \left[ c_1 \log \frac{c_2}{\delta \Delta_i^\zeta} + \frac{\rho\epsilon}{2} \right] + \max_{i \in C \cap \mathcal{N}} \left\{ \frac{2}{(\Delta_i^\zeta)^2} \left[ c_1 \log \frac{c_2}{\delta \Delta_i^\zeta} + \frac{\rho\epsilon}{2} \right] \right\} \right\rceil \quad (70)$$

where, summing it over all the components $C \in \mathcal{C}$, gives us the sample complexity for GRUB while considering graph $G$.

Any subgraph $D$ of graph $G$ satisfies,

$$\langle \boldsymbol{\mu}, L_G \boldsymbol{\mu} \rangle \leq \epsilon \Rightarrow \langle \boldsymbol{\mu}, L_D \boldsymbol{\mu} \rangle \leq \epsilon \quad (71)$$

As seen in Definition G.1, the influence factor is instrumental in deciding the competitive and non-competitive sets, which further dictates the sample complexity bounds. Further, notice from Lemma I.8 that the influence factor $\mathfrak{I}(i, D)$ is not monotonic when considering subgraph $D$ of graph $G$. Hence considering a subgraph of $G$ could potentially increase the number of non-competitive arms and provide us with a tighter bound on the performance for GRUB.

Hence $T_{\text{sufficient}}$ can be made tighter by considering the minimum value over the entire set of subgraphs $\mathcal{G}$. $\qquad\qquad\qquad\qquad\qquad\qquad\qquad\qquad\qquad\qquad\qquad\qquad\qquad\qquad\qquad\qquad\square$

Note that, as in the case of GRUB, the $\zeta$-GRUB algorithm's performance *automatically* adapts to the best possible subgraph in $\mathcal{G}$.

## H   The Incomparability of the Graph Bandits problem with Linear Bandits

In this section, we demonstrate an example graph bandit problem that is cast as a linear bandit to reveal the incomparability of these frameworks. A typical linear bandit problem is defined as follows: Consider an $n$-armed linear bandit problem, each arm $i \in [n]$ is associated with a feature vector $\mathbf{x}_i \in \mathbb{R}^d$, where $d$ can be lower than $n$. In each round $t$, the learner chooses an action $\mathbf{a}_t = \mathbf{x}_i$ for some $i \in [n]$ and observes the reward $y_t = \langle \mathbf{a}_t, \boldsymbol{\theta} \rangle + \eta_t$, where $\boldsymbol{\theta} \in \mathbb{R}^d$ is an unknown parameter and the $\eta_t$ is a subgaussian random noise with $\sigma^2$ variance. Denote the arm with the best mean reward with $i^*$, i.e. $i^* = \arg\max_{i \in [n]} \langle \mathbf{x}_i, \boldsymbol{\theta} \rangle$. The goal of the learner is to to output the index of the arm $i^*$ with probability $1 - \delta$, $\delta > 0$ in as few samples as possible.

Firstly, a $n$-armed bandit problem without any graph can be easily seen as linear bandits by associating the canonical basis for $\mathbb{R}^n$ $\{\mathbf{e}_i\}_{i=1}^n$ as the feature vectors and the mean vector $\boldsymbol{\mu} \in \mathbb{R}^n$ as the unknown reward vector. This provides up with the mean reward function for arm $i \in [n]$ as $\langle \mathbf{e}_i, \boldsymbol{\mu} \rangle = \mu_i$.

In order to cast the graph bandit problem in a linear bandit framework, we need to associate every arm index $i$ with a feature vector $\mathbf{x}_i$ and identify the unknown feature vector $\boldsymbol{\theta}$ for the problem. We achieve this by modifying the feature vectors $\{\mathbf{e}_i\}_{i=1}^n$ and the reward vector $\boldsymbol{\mu}$ based on the graph Laplacian $L_G$.

Following is the information available at hand in the current graph bandit problem: we are provided with an $n$-armed bandit with an unknown mean vector $\boldsymbol{\mu}$ smooth on a graph $G$, i.e. $\langle \boldsymbol{\mu}, L_G \boldsymbol{\mu} \rangle \leq \epsilon$. For this toy problem, we consider the graph $G$ to be connected.

Let $\{\boldsymbol{\nu}_i\}_{i=1}^n$ and $0 = \lambda_1 < \cdots < \lambda_n$ denote the eigenvectors and eigenvalues of the Laplacian $L_G$ respectively. It can be easily seen that $\boldsymbol{\mu} = \sum_{i=1}^n a_i \boldsymbol{\nu}_i$ for some $a_i \geq 0 \ \ \forall i \in [n]$. The reward function of arm $j$ is

$$\langle \mathbf{e}_j, \boldsymbol{\mu} \rangle = \sum_{i=1}^n a_i \langle \mathbf{e}_j, \boldsymbol{\nu}_i \rangle = a_1 + \sum_{i=2}^n a_i \langle \mathbf{e}_j, \boldsymbol{\nu}_i \rangle$$

the second equality follows from the properties of graph Laplacian we know that $\boldsymbol{\nu}_1 = \mathbb{1}_n$, is the only eigenvector associated to 0 eigenvalue in a connected graph.

Without loss of generality we can assume $a_1 = 0$ as $a_1$ does not depend on the arm index $j$. Notice that letting $a_1 = 0$ is equivalent to having $\sum_{i=1}^n \mu_i = 0$. Also, the graph constraint can be rewritten as follows:

$$\langle \boldsymbol{\mu}, L_G \boldsymbol{\mu} \rangle \leq \epsilon \Rightarrow \sum_{i=1}^n \lambda_i a_i^2 = \langle \boldsymbol{\theta}, \boldsymbol{\theta} \rangle = \|\boldsymbol{\theta}\|_2^2 \leq \epsilon$$

where $\boldsymbol{\theta} = (\sqrt{\lambda_1} a_1, \ldots, \sqrt{\lambda_n} a_n)$.

Using the above we can cast the graph bandit problem as the linear bandit problem with the mean reward function of arm $j$ expressed as

$$\langle \mathbf{e}_j, \boldsymbol{\mu} \rangle = \sum_{i=2}^n \frac{\theta_i}{\sqrt{\lambda_i}} \langle \mathbf{e}_j, \nu_i \rangle = \langle \mathbf{x}_j, \boldsymbol{\theta} \rangle$$

Hence, the new linear bandit problem is such that the set of arms is $\{\mathbf{x}_j\}_{j=1}^n$, the unknown parameter is a vector $\boldsymbol{\theta}$, the expected reward of an arm is $\langle \mathbf{x}_j, \boldsymbol{\theta} \rangle$ and the unknown parameter satisfies the constraint $\|\boldsymbol{\theta}\|_2^2 \leq \epsilon$.

We discuss below the drawbacks of casting a graph bandit problem into a linear bandit framework:

- The original best-arm identification is an $n$-armed problem and the recasted linear bandit problem still has feature vectors with dimensionality $n$ and hence no low-dimensional

benefit of linear bandits is completely lost. Having a performance bound for any algorithm for linear bandits which scales in $n$, the number of arms gives us no additional advantage.

- The above conversion to linear bandit setup only works when the graph $G$ is connected. Recasting problem setup with disconnected components require assumption of $\sum_{i \in C} \mu_i = 0$ on individual connected components, which is unrealistic. The results of GRUB holds with or without this assumption.

- Consider the corner case of $\epsilon = 0$, the linear bandit problem setup derived becomes that of $\arg \max_i \langle \mathbf{x}_i, \boldsymbol{\theta} \rangle$ such that $\|\boldsymbol{\theta}\| \leq 0$ which is only possible if $\|\boldsymbol{\theta}\| = 0$ and in this case we can observe two interesting facts:

  - If the graph $G$ is completely connected then the problem is trivial, since

  $$\epsilon = 0 \Rightarrow \langle \boldsymbol{\mu}, L_G \boldsymbol{\mu} \rangle = 0 \Rightarrow (\mu_i - \mu_j)^2 = 0 \ \forall i, j \in [n], i \neq j$$

  This implies all arms are equal and optimal and the solution is trivial. Here the mean reward function of all arms $i$ is $\langle \mathbf{x}_i, \boldsymbol{\theta} \rangle = 0$ since $\theta = 0$ and hence gives the correct output (any arm $i$).

  - Suppose graph $G$ has two connected components $C_1, C_2$, where $C_k$ indicates the arm indices in the connected component $k$. Further assume that $\mu_i = 1 \ \forall i \in C_1, \mu_i = -1 \ \forall i \in C_2$. Considering the case of $\epsilon = 0$ here gives us the following :

  $$\epsilon = 0 \Rightarrow \langle \boldsymbol{\mu}, L_G \boldsymbol{\mu} \rangle = 0 \Rightarrow (\mu_i - \mu_j)^2 = 0 \ \forall i \neq j, i, j \in C_k, k = 1, 2$$

  Here the mean reward function of all arms $i$ is $\langle \mathbf{x}_i, \boldsymbol{\theta} \rangle = 0$ since $\theta = 0$ but this is incorrect since not all arms are optimal.

  Our graph bandit setup and the performance of GRUB is independent of all of these drawbacks and provides us with a better sample complexity than vanilla best arm identification algorithms.

# I  Supporting Results

This appendix is devoted to providing supporting results for many of the theorems and lemmas in the paper.

## I.1  Notation and Definition

Let $\{t_i(T)\}_{i=1}^n$ (denoted as $\{t_i\}_{i=1}^n$ for ease of reading) indicate the number of plays of each arm until time $T$. Let $X \in \mathbb{R}^{n \times n}$ be a matrix, then $\{\lambda_i(X)\}_{i=1}^n$ indicate the eigenvalues of matrix $X$ in an increasing order.

Let $N(\boldsymbol{\pi}_T) = \sum_{t=1}^T \mathbf{e}_{\pi_t} \mathbf{e}_{\pi_t}^T$ be the diagonal counting matrix. Note that $N(\boldsymbol{\pi}_T)$ can be written as $N(\{t_i\}_{i=1}^n)$ since the diagonal counting matrix only depends on the number of plays of each arm, rather than the each sampling sequence $\boldsymbol{\pi}_T$.

We next establish some properties of the influence function $\mathfrak{I}$.

**Lemma I.1.** *Let $D$ be an arbitrary graph with $n$ nodes and let $\{t_i\}_{i=1}^n$ be the number of times all arms are sampled till time $T$. For each node $j \in [n]$, the following are equivalent:*

$$\frac{1}{\mathfrak{I}(j, D)} = \max_{\sum_{i \in D_j, i \neq j} t_i = T} \left\{ [K(i, D)]_{jj} \right\} \quad (A)$$

$$= \max_{k \in D_j, \sum_{i \in D_j, i \neq j} t_i = T} \left\{ [V_j(\{t_i\}_{i \in D_j}, D)^{-1}]_{jj} - [V_j(\{t_i\}_{i \in D_j}, D)^{-1}]_{kk} \right\} \quad (B)$$

$$= \max_{\sum_{i \in D_j, i \neq j} t_i = T} \left\{ [V_j(\{t_i\}_{i \in D_j}, D)^{-1}]_{jj} - \min_{k \in D_j} [V_j(\{t_i\}_{i \in D_j}, D)^{-1}]_{kk} \right\} \quad (C)$$

$$= \max_{\sum_{i \in D_j, i \neq j} t_i = T} \left\{ [V_j(\{t_i\}_{i \in D_j}, D)^{-1}]_{jj} - \frac{1}{T} \right\} \quad (D) \tag{72}$$

*where $K(i, D)$ be defined as in Definition 4.2*

*Proof.* Let $f(\cdot, \cdot)$ denote the following:

$$f(i, D) = \max_{\sum_{i \in D_j, i \neq j} t_i = T} \{[K(i, D)]_{jj}\}$$

We prove the rest by showing equivalence between $(A), (B), (C)$ and $(D)$.

- $(A) \Leftrightarrow (D)$ : A simple extension of Lemma I.3 to the case of disconnected clustered graph $D, \forall \boldsymbol{\pi}_T \in \mathcal{U}(T, D_j)$ we obtain,

$$V_j(\boldsymbol{\pi}_T, D)^{-1} = \frac{1}{T} \mathbb{1}\mathbb{1}^T + K(\pi_1, D) \tag{73}$$

  where $K(\pi_1, D)$ is as defined in Definition 4.2. Thus, we have the equivalence by explicitly writing the diagonal element of eq (73),

$$[V_j(\boldsymbol{\pi}_T, D)^{-1}]_{jj} - \frac{1}{T} = [K(\pi_1, D)]_{jj} \tag{74}$$

  Hence we have the equivalence as,

$$f(i, D) = \max_{\sum_{i \in D_j, i \neq j} t_i = T} \left\{ [V_j(\{t_i\}_{i \in D_j}, D)^{-1}]_{jj} - \frac{1}{T} \right\} \tag{75}$$

- $(C) \Leftrightarrow (D)$ : Let $\{t_i^*\}_{i \in D_j}$ denote the following:

$$\{t_i^*(j)\}_{i \in D_j} \in \operatorname*{arg\,max}_{\sum_{i \in D_j, i \neq j} t_i = T} \left\{ [V_j(\{t_i\}_{i \in D_j}, D)]_{jj}^{-1} - \frac{1}{T} \right\} \tag{76}$$

  From Lemma I.2, the optimal $\{t_i^*(j)\}_{i \in D_j}$ occurs in $\mathcal{U}(j, T)$, i.e. $\exists \{t_i^*(j)\}_{i \in D_j}$ such that $t_l^*(j) = T$ and $t_k^*(j) = 0 \ \forall k \neq l$ for some $l \in D_j$. Further by Lemma I.4,

$$\min_{k \in D_j} [V_j(\{t_i\}_{i \in D_j}, D)^{-1}]_{kk} = \frac{1}{T} \tag{77}$$

  Hence $\{t_i^*(j)\}_{i \in D_j}$ is also a solution for the following problem:

$$\{t_i^*(j)\}_{i \in D_j} \in \operatorname*{arg\,max}_{\sum_{i \in D_j, i \neq j} t_i = T} \left\{ [V_j(\{t_i\}_{i \in D_j}, D)]_{jj}^{-1} \right.$$
$$\left. - \min_{k \in D_j} [V_j(\{t_i\}_{i \in D_j}, D)^{-1}]_{kk} \right\} \tag{78}$$

  Hence we can conlcude that,

$$f(i, D) = \max_{\sum_{i \in D_j, i \neq j} t_i = T} \left\{ [V_j(\{t_i\}_{i \in D_j}, D)]_{jj}^{-1} \right.$$
$$\left. - \min_{k \in D_j} [V_j(\{t_i\}_{i \in D_j}, D)^{-1}]_{kk} \right\} \tag{79}$$

- $(B) \Leftrightarrow (C)$ :

  Note that $\max_{k \in D_j, \sum_{i \in D_j, i \neq j} t_i = T} [V_j(\{t_i\}_{i \in D_j}, D)^{-1}]_{jj}$ does not depend on arm node index $k \in D_j$. Hence, the equivalence follows.

The resistance distance $r(i, j)$ Definition 4.1 is independent of $\delta$ for all $i, j \in [n]$ (The addition of diagonal elements and subtraction of off diagonal elements removes the dependence on $\delta$ [5]).

Note that $V_T = N_T + \rho L_G$, hence $V_T^{-1}$ gives the psuedo-inverse of the Laplacian matrix for graph $G$. We show in Lemma I.2 that the matrix $R$ (denoting as $R(\delta)$ to explicitly show dependence on $\delta$) linked with $V_T^{-1}$ is independent of number of samples $T$. Since both matrix $R$ and $V_T$ are psuedo-inverse of the Laplacian $L_G$. Thus we can conclude the following :

$$\lim_{\delta \to 0} [R(\delta)]_{ij} - \frac{1}{\delta} = \lim_{T \to 0} [V(\{t_i\}_{i=1}^n, G)^{-1}]_{ij} - \frac{1}{T} \tag{80}$$

where $T \to 0$ implies $t_i \to 0 \quad \forall i \in [n]$. Further,

$$\lim_{\delta \to 0} R(\delta)_{ii} + R(\delta)_{jj} - R(\delta)_{ij} - R(\delta)_{ji}$$

$$= \lim_{T \to 0} [V(\{t_i\}_{i=1}^n, G)^{-1}]_{ii} + [V(\{t_i\}_{i=1}^n, G)^{-1}]_{jj}$$

$$- [V(\{t_i\}_{i=1}^n, G)^{-1}]_{ij} - [V(\{t_i\}_{i=1}^n, G)^{-1}]_{ji} \tag{81}$$

where $T \to 0$ implies $t_i \to 0 \quad \forall i \in [n]$.

Since the equation (81) holds for $t_i \to 0$ for all $i \in [n]$, computing the value of limit for one trajectory should suffice for finding the value of the limit. Thereby, we provide an alternate equation for obtaining the resistance distance $r(i,j)$ by

$$r(i,j) = [K(\pi_1 = i, D)]_{jj} \tag{82}$$

Note that $[K(\pi_1 = i, D)]_{ii} = [K(\pi_1 = i, D)]_{ij} = [K(\pi_1 = i, D)]_{ji} = 0$ from Lemma I.3). Thus we can say from Definition 4.2,

$$f(i, D) = \frac{1}{\mathfrak{I}(j, D)}$$

Hence proved.

$\square$

**Lemma I.2.** *Let $D$ be a given graph with $n$ nodes. For every node $j \in D$, let $\{t_i^*(j)\}_{i \in D_j}$ denote the following:*

$$\{t_i^*(j)\}_{i \in D_j} \in \underset{\sum_{i \in D_j, i \neq j} t_i = T}{\arg \max} \left\{ [V_j(\{t_i\}_{i \in D_j}, D)]_{jj}^{-1} - \frac{1}{T} \right\} \tag{83}$$

*Then $\exists \{t_i^*(j)\}_{i \in D_j}, l \in D_j$ such that $t_l^*(j) = T$ and $t_k^*(j) = 0 \quad \forall k \neq l$.*

*Proof.* To simplify our proof, let graph $D$ be connected. The proof for the case of disconnected components is an extension of the connected graph case, by analysing each individual connected component together.

If graph $D$ is connected then $D_i = D$. For the rest of the proof we sometimes denote $V(\boldsymbol{\pi}_T, D)$ as $V(\{t_i\}_{i=1}^n, D)$ to make it more context relevant.

Let $g: \mathbb{R}^n \to \mathbb{R}^{n \times n}$ be a partial function of $V(\boldsymbol{\pi}_T, D)$ as follows:

$$g(\{t_i\}_{i=1}^n) = V(\{t_i\}_{i=1}^n, D) \tag{84}$$

For all $i \in [n]$, let $t_i = \alpha_i T$ such that $\sum_{i=1}^n \alpha_i = 1$. Then we can say that,

$$g(\{t_i\}_{i=1}^n) = g(\{\alpha_i T\}_{i=1}^n)$$

$$= \sum_{i=1}^n \alpha_i g(\{0, 0, \ldots t_i = T, \ldots 0\}) \tag{85}$$

Using convexity of matrix invertibility [42] $V(\boldsymbol{\pi}_T, G)^{-1}$ satisfies,

$$g(\{t_i\}_{i=1}^n)^{-1} \preceq \sum_{i=1}^n \alpha_i g(\{0, 0, \ldots t_i = T, \ldots 0\})^{-1} \tag{86}$$

Hence $g(\cdot)^{-1}$ is a convex function. Since we have the restriction as $\sum_{i=1, i \neq j}^n t_i = T$. We can say that,

$$\underset{\sum_{i \in D_j, i \neq j} t_i = T}{\arg \max} \left\{ [V(\{t_i\}_{i=1}^n, D)]_{jj}^{-1} - \frac{1}{\sum_{i=1}^n t_i} \right\}$$

$$= \underset{\sum_{i \in D_j, i \neq j} t_i = T}{\arg \max} [V(\{t_i\}_{i=1}^n, D)]_{jj}^{-1}$$

$$= \underset{\sum_{i \in D_j, i \neq j} t_i = T}{\arg \max} \langle \mathbf{e}_j, [V(\{t_i\}_{i=1}^n, D)]^{-1} \mathbf{e}_j \rangle$$

$$= \underset{\sum_{i \in D_j, i \neq j} t_i = T}{\arg \max} \langle \mathbf{e}_j, g(\{t_i\}_{i=1}^n)^{-1} \mathbf{e}_j \rangle \tag{87}$$

Since $g(\cdot)^{-1}$ is convex, for a convex function the maximization over a simplex happens at one of the vertices. Hence the max happens when $t_i = T$ and $t_k = 0 \quad \forall k \neq i$.

Hence proved. $\qquad\square$

**Lemma I.3.** *Let $G$ be a given connected graph of $n$ nodes and $t_i$ be the number of samples of each arm $i$. Then $\forall \boldsymbol{\pi}_T \in \mathcal{U}(T)$,*

$$V(\boldsymbol{\pi}_T, G)^{-1} = \frac{1}{T} \mathbb{1}\mathbb{1}^T + K(\pi_1, G) \tag{88}$$

*where, $\mathbb{1} \in \mathbb{R}^n$ is a vector or all ones and $K(\pi_1, G) \in \mathbb{R}^{n \times n}$ is the matrix defined in Definition 4.2.*

*Proof.* Let $I$ be an identity matrix of dimension $n \times n$. We prove the result by showing that, $\forall \boldsymbol{\pi}_T \in \mathcal{U}(T), V(\boldsymbol{\pi}_T, G)^{-1} V(\boldsymbol{\pi}_T, G) = I$,

$$
V(\boldsymbol{\pi}_T, G)^{-1} V(\boldsymbol{\pi}_T, G)
$$
$$
= \left( \frac{1}{T} \mathbb{1}\mathbb{1}^T + K(\pi_1, G) \right) \left( \sum_{t=1}^{T} \mathbf{e}_{\pi_t} \mathbf{e}_{\pi_t}^T + \rho L_G \right)
$$
$$
= \left( \frac{1}{T} \mathbb{1}\mathbb{1}^T + K(\pi_1, G) \right) \left( T \mathbf{e}_{\pi_1} \mathbf{e}_{\pi_1}^T + \rho L_G \right)
$$
$$
= \mathbb{1}\mathbf{e}_{\pi_1}^T + T K(\pi_1, G) \mathbf{e}_{\pi_1} \mathbf{e}_{\pi_1}^T + \rho K(\pi_1, G) L_G \tag{89}
$$

From Definition 4.2, $K(\pi_1, G) \mathbf{e}_{\pi_1} \mathbf{e}_{\pi_1}^T = 0$ and $\mathbb{1}\mathbf{e}_{\pi_1}^T + \rho K(\pi_1, G) L_G = I$ implying that $V(\boldsymbol{\pi}_T, G)^{-1} V(\boldsymbol{\pi}_T, G) = I$.

Hence proved. $\qquad\square$

**Lemma I.4.** *Let $G$ be any connected graph and $\boldsymbol{\pi}_T \in \mathcal{U}(T, G)$. Then,*

$$\min_{j \in [n]} \{ [V(\boldsymbol{\pi}_T, G)^{-1}]_{jj} \} = \frac{1}{T} \tag{90}$$

*Proof.* From Definition 4.2, $K(\pi_1, G)$ satisfies

$$K(\pi_1, G) L_G = \frac{1}{\rho} \left( I - \mathbb{1}\mathbf{e}_{\pi_1}^T \right)$$

Observe that $\mathbb{1}\mathbf{e}_i^T$ is a rank 1 matrix with eigenvalue 1 and eigenvector $\mathbf{e}_i$ and Identity matrix $I$ is of rank $n$ with all eigenvalues 1 and eigenvectors $\{\mathbf{e}_i\}_{i=1}^n$. Hence $\left( I - \mathbb{1}\mathbf{e}_{\pi_1}^T \right)$ is a rank $n-1$ matrix with rest nonzero eigenvalues as 1. Since the graph $G$ is connected, $\lambda_1(L_G) = 0$ and $\lambda_2(L_G) > 0$. The eigenvector corresponding to $\lambda_1(L_G)$ is $\mathbb{1}$, the all 1 vector.

Given $\rho > 0$, we can conclude,

$$K(\pi_1, G) L_G \succeq 0 \quad \text{s.t. } \operatorname{rank}(K(\pi_1, G) L_G) = n - 1 \tag{91}$$

Hence, in order to satisfy eq. (91), $K(\pi_1, G) \succeq 0$ and $\operatorname{rank}(K(\pi_1, G)) \geq n - 1$. By lower bounds on Rayleigh quotient we can conclude,

$$\langle \mathbf{e}_j, K(\pi_1, G) \mathbf{e}_j \rangle = [K(\pi_1, G)]_{jj} \geq 0 \quad \forall j \in [n] \tag{92}$$

From Lemma I.3, $[K(\pi_1, G)]_{jj} = [V(\boldsymbol{\pi}_T, G)^{-1}]_{jj} - \frac{1}{T}$ implying that $[V(\boldsymbol{\pi}_T, G)^{-1}]_{jj} \geq \frac{1}{T}$. From Definition 4.2 it can be seen that $[K(\pi_1, G)]_{\pi_1 \pi_1} = 0$ and hence $[V(\boldsymbol{\pi}_T, G)^{-1}]_{\pi_1 \pi_1} = \frac{1}{T}$ which concludes the proof. $\qquad\square$

**Lemma I.5.** *Given a connected graph $G$, the following bound holds for all the diagonal entries of $[V(\boldsymbol{\pi}_T, G)^{-1}]_{ii}$ for $i \in [n]$:*

$$[V(\boldsymbol{\pi}_T, G)^{-1}]_{ii} \leq \mathbb{1}(t_i = 0) \left( \frac{1}{\rho \mathfrak{I}(i, \mathcal{G})} + \frac{1}{T} \right) + \mathbb{1}(t_i > 0) \max \left\{ \frac{1}{t_i + \frac{\rho \mathfrak{I}(i, \mathcal{G})}{2}}, \frac{1}{t_i + \frac{T}{2}} \right\} \tag{93}$$

*Proof.* From Definition 4.2 of $\mathfrak{I}(\cdot, \mathcal{G})$ and Lemma I.1, Breaking the lemma statement into cases:

- **Unsampled Arms :** From Lemma I.1

$$\frac{1}{\mathfrak{I}(j, G)} = \max_{\sum_{i \in G_j, i \neq j} t_i = T} \left\{ [V_j(\{t_i\}_{i \in G_j}, G)^{-1}]_{jj} - \frac{1}{T} \right\} \quad \forall j \in [n] \tag{94}$$

Thus for any unsampled arm $j$,

$$[V(\boldsymbol{\pi}_T, G)]_{jj}^{-1} \leq \left( \frac{1}{\mathfrak{I}(j, G)} + \frac{1}{T} \right) \tag{95}$$

- **Sampled Arms :** Since the matrix $V(\boldsymbol{\pi}_T, G)$ depends only on the final sampling distribution $\{t_i\}_{i=1}^n$ rather than the sampling path $\boldsymbol{\pi}_T$. Consider a sampling path such that $\pi_t \neq j$ for $t \leq T - t_j$ and $\pi_t = j$ for $T - t_j \leq t \leq T$.

Assuming such a sampling path $\boldsymbol{\pi}_T$, after $\boldsymbol{\pi}_{T-t_j}$ samples,

$$[V(\boldsymbol{\pi}_{T-t_j}, G)^{-1}]_{jj} \leq \frac{1}{T} + \frac{1}{\mathfrak{I}(j, G)} \tag{96}$$

Then by the Sherman-Morrison rank 1 update identity[9],

$$\frac{1}{[V(\boldsymbol{\pi}_T, G)^{-1}]_{jj}} = \frac{1}{[V(\boldsymbol{\pi}_{T-t_j}, G)^{-1}]_{jj}} + t_j$$

$$[V(\boldsymbol{\pi}_T, G)^{-1}]_{jj} = \frac{1}{t_j + \frac{1}{[V(\boldsymbol{\pi}_{T-t_j}, G)^{-1}]_{jj}}}$$

$$\leq \frac{1}{t_j + \frac{1}{\left( \frac{1}{\mathfrak{I}(j, G)} + \frac{1}{T-t_j} \right)}}$$

Hence we have the bound on $[V(\boldsymbol{\pi}_T, G)^{-1}]_{jj}$ as follows:

$$[V(\boldsymbol{\pi}_T, G)^{-1}]_{jj} \leq \max \left\{ \frac{1}{t_j + \frac{\mathfrak{I}(j, G)}{2}}, \frac{1}{t_j + \frac{T-t_j}{2}} \right\}$$

$$\tag{97}$$

Hence proved. $\qquad\square$

**Lemma I.6.** *Let $D$ be a graph with $n$ nodes and $k$ disconnected components. If each of the connected components $\{\mathcal{C}_i(D)\}_{i=1}^k$ is a complete graph then $\forall \, j \in [n]$,*

$$\mathfrak{I}(j, D) = \frac{|\mathcal{C}(j, D)|}{2} \tag{98}$$

*Proof.* Let $D$ be a complete graph (k = 1), $\boldsymbol{\pi}_T \in \mathcal{U}(T)$ and $\rho = 1$. Then,

$$V(\boldsymbol{\pi}_T, G)^{-1} = \frac{1}{T} \mathbb{1}\mathbb{1}^T + K \tag{99}$$

where $\mathbb{1} \in \mathbb{R}^n$ is a vector or all ones and $K \in \mathbb{R}^{n \times n}$ is a matrix given by,

$$K_{\pi_1 \pi_1} = 0, \quad K_{jj} = \frac{2}{n} \quad \forall j \in [n]/\{\pi_1\}$$

$$K_{k\pi_1} = 0, \quad K_{\pi_1 j} = 0, \quad K_{jk} = \frac{1}{n} \quad \forall j, k \in [n]/\{\pi_1\}, \; j \neq k$$

The form of $V(\boldsymbol{\pi}_T, G)^{-1}$ in eq.(99) can be verified by $V(\boldsymbol{\pi}_T, G)^{-1} V(\boldsymbol{\pi}_T, G) = I$.

The final statement of the lemma can be obtained by considering this analysis to just the nodes within a connected component of a diconnected graph $G$ and Lemma I.1. $\qquad\square$

---

[9]Hager, W. (1989). Updating the Inverse of a Matrix. SIAM Rev., 31, 221-239.

**Lemma I.7.** *Let $D$ be a graph with $n$ nodes and $k$ disconnected components. If each of the connected components $\{\mathcal{C}_i(D)\}_{i=1}^k$ is a line graph then $\forall\, j \in [n]$,*

$$\Im(j, D) > \frac{1}{|\mathcal{C}(j, D)|} \tag{100}$$

*Proof.* Let $D$ be a complete graph (k = 1), $\boldsymbol{\pi}_T \in \mathcal{U}(T)$ and $\rho = 1$. Then,

$$V(\boldsymbol{\pi}_T, G)^{-1} = \frac{1}{T}\mathbb{1}\mathbb{1}^T + K \tag{101}$$

where $\mathbb{1} \in \mathbb{R}^n$ is a vector or all ones and $K \in \mathbb{R}^{n \times n}$ is a matrix given by,

$$K_{\pi_1\pi_1} = 0, \;\; K_{jj} = d(\pi_1, j) \;\; \forall j \in [n]/\{\pi_1\},$$
$$K_{k\pi_1} = 0, \;\; K_{\pi_1 j} = 0,$$
$$K_{jk} = \min\{d(\pi_1, j), d(\pi_1, k)\} \;\; \forall j, k \in [n]/\{\pi_1\}, \; j \neq k$$

The form of $V(\boldsymbol{\pi}_T, G)^{-1}$ in eq.(101) can be verified by $V(\boldsymbol{\pi}_T, G)^{-1}V(\boldsymbol{\pi}_T, G) = I$.

The final statement of the lemma can be obtained by considering this analysis to just the nodes within a connected component of a diconnected graph $G$ and Lemma I.1. $\qquad\square$

**Lemma I.8.** *Let $A = ([n], E)$ be any graph and let $e \in E$ be an edge of graph $A$. Let $B = ([n], E - \{e\})$ be a subgraph of $A$ with one edge removed. Then the following holds for all non-isolated nodes $i$ in $B$:*

- *If $|\mathcal{C}(A)| = |\mathcal{C}(B)|$,*

$$\Im(i, A) \geq \Im(i, B)$$

- *If $|\mathcal{C}(A)| < |\mathcal{C}(B)|$,*

$$\Im(i, A) \leq \Im(i, B)$$

*Proof.* From Lemma I.1, for any graph $D$, $\Im(\cdot, \cdot)$ satisfies,

$$\frac{1}{\Im(j, D)} = \max_{k \in D_j, \sum_{i \in D_j} t_i = T} \left\{ [V_j(\{t_i\}_{i \in D_j}, D)^{-1}]_{jj} \right.$$
$$\left. - [V_j(\{t_i\}_{i \in D_j}, D)^{-1}]_{kk} \right\} \quad \forall j \in [n] \tag{102}$$

**Case I :** $|\mathcal{C}(A)| = |\mathcal{C}(B)|$

The edge set of $B$ is smaller than edge set of $A$. Hence, from Lemma $\Im(i, A) \geq \Im(i, B)$

**Case II :** $\mathcal{C}(A) < \mathcal{C}(B)$ In this case, $|B_i| \leq |A_i|$. Hence the $\max$ is over a smaller set of options, we can conclude that $\Im(i, A) \leq \Im(i, B)$. Hence proved. $\qquad\square$

Given a graph $D$, we define a class of sampling policies $\mathcal{U}(T, D)$ as follows,

**Definition I.9.** *Let $\mathcal{U}(T, D)$ denote the set of sampling policies,*

$$\mathcal{U}(T, D) = \{\boldsymbol{\pi}_T |\; \exists l \in D \;\text{ s.t. } \pi_t = l \;\; \forall t \leq T\}$$

**Lemma I.10.** *Let $G$ be the given graph and sampling policy $\boldsymbol{\pi}_T$ has been played for $T$ time steps, then $V_T$ satisfies the following structure,*

$$V(\boldsymbol{\pi}_T, D) = diag([V_1, V_2, \ldots, V_{k(G)}]) \tag{103}$$

*where $V_i$ depends on the connected component $C_i \in \mathcal{C}_D$ of the graph and the number of samples of the arms within the connected component $\{t_j\}_{j \in C_i}$.*

*Proof.* Rewriting the definition of $V(\boldsymbol{\pi}_T, D)$,

$$V(\boldsymbol{\pi}_T, D) \triangleq \sum_{t=1}^T \mathbf{e}_{\pi_t}\mathbf{e}_{\pi_t}^\top + \rho L_D$$
$$= N(\{t_i\}_{i=1}^n) + L_D \tag{104}$$

Both component matrices $N(\{t_i\}_{i=1}^n)$ (diagonal matrix) and $L_D$ (Laplacian matrix of a graph) adhere to a block diagonal structure and hence $V(\boldsymbol{\pi}_T, D)$ matrix also adheres to a block diagonal structure analogous to $L_D$. The block diagonal structure in $L_D$ is dictated by connected components of graph $D$. $\qquad\square$

The following lemma establishes the invertibility of $V(\boldsymbol{\pi}_T, G)$ for a connected graph and $T > 1$:

**Lemma I.11.** *For a connected graph $G$, $V(\boldsymbol{\pi}_1, G)$ is invertible, but $V(\boldsymbol{\pi}_0, G)$ is not invertible.*

*Proof.* Since the graph $G$ is connected, $\lambda_1(L_G) = 0$ and $\lambda_2(L_G) > 0$. The eigenvector corresponding to $\lambda_1(L_G)$ is $\mathbb{1}$, the all 1 vector. At time $T = 0$, $V(\boldsymbol{\pi}_T, G) = L_G$ and hence $V(\boldsymbol{\pi}_T, G)$ is positive semi-definite matrix with one zero eigenvalues.

Let arm $i$ be pulled at $T = 1$, i.e. $\pi_1 = i$, then the corresponding counting matrix is a positive semi definite matrix of rank one with the eigen value $\lambda_n(N) = 1$ for the eigenvector $e_i$.

Observe that $\mathbf{e}_i^T \mathbb{1} > 0$. Also, $N_T$ and $L_G$ are positive semi-definite matrices with ranks 1 and $n - 1$ respectively. The subspace without information (corresponding to the direction of zero eigenvalue) for matrix $L_G$ is now provided by $N(\boldsymbol{\pi}_1)$ and hence $\lambda_{\min}(V(\boldsymbol{\pi}_1, G)) > 0$ making it invertible. $\quad\square$

**Lemma I.12.** *Let $G = ([n], E_G, A), H = ([n], E_H, A)$ are two graphs with $n$ nodes such that $E_G \supseteq E_H$. Then, assuming invertibility of $[V(G,T)^{-1}]$ and $[V(H,T)^{-1}]$,*

$$[t_{eff,i}]_G \geq [t_{eff,i}]_H \quad \forall i \in [n], T > k(G) \tag{105}$$

*where $\forall i \in [n]$, $[t_{eff,i}]_G, [t_{eff,i}]_H$ indicates the effective samples with graph $G$ and $H$ respectively.*

*Proof.* Given graphs $G = ([n], E_G), H = ([n], E_H)$ satisfy $E_G \supseteq E_H$.

The quadratic form of Laplacian for the graph $G, H$ is given by,

$$\mathbf{x}L_G\mathbf{x} = \sum_{(i,j)\in E_G} (x_i - x_j)^2$$

$$\mathbf{x}L_H\mathbf{x} = \sum_{(i,j)\in E_H} (x_i - x_j)^2$$

Since $E_G \supseteq E_H$,

$$\mathbf{x}L_G\mathbf{x} \geq \mathbf{x}L_H\mathbf{x} \quad \forall\, \mathbf{x} \in \mathbb{R}^n$$
$$\Rightarrow L_G \succeq L_H$$

Further, provided a sampling policy $\boldsymbol{\pi}_T$, we can say that,

$$V(\boldsymbol{\pi}_T, G) \succeq V(\boldsymbol{\pi}_T, H)$$

For the number of samples $T$ sufficient to ensure invertibility of $V(\boldsymbol{\pi}_T, H)$, we have

$$V(\boldsymbol{\pi}_T, G)^{-1} \preceq V(\boldsymbol{\pi}_T, H)^{-1}$$
$$\mathbf{x}^T V(\boldsymbol{\pi}_T, G)^{-1}\mathbf{x} \leq \mathbf{x}^T V(\boldsymbol{\pi}_T H)^{-1}\mathbf{x} \quad \forall \mathbf{x} \in \mathbb{R}^n$$
$$[V(\boldsymbol{\pi}_T, G)^{-1}]_{ii} \leq [V(\boldsymbol{\pi}_T, H)^{-1}]_{ii} \quad \text{(taking } \mathbf{x} = \mathbf{e}_i)$$
$$\frac{1}{[V(\boldsymbol{\pi}_T, G)^{-1}]_{ii}} \geq \frac{1}{[V(\boldsymbol{\pi}_T, H)^{-1}]_{ii}}$$

Hence from the definition of effective samples 3.1, it is clear that for any $i \in [n]$,

$$[t_{\text{eff},i}]_G \geq [t_{\text{eff},i}]_H \tag{106}$$

Hence proved. $\qquad\square$

**Lemma I.13.** *Let effective samples $t_{eff,i}$ be as is defined in Definition 3.1 and let $\boldsymbol{\pi}_T$ denote a cyclic sampling policy for $T > k(G)$, then the infinite sum $\sum_{T=k(G)+1}^{\infty} t_{eff,i}^{-2}$ is bounded. In fact,*

$$\sum_{T=k(G)+1}^{\infty} t_{eff,i}^{-2} < n\left(\frac{2(n-1)}{\rho}\right)^2 + \frac{n\pi^2}{6} \tag{107}$$

*Proof.* We first prove the lemma statement for connected graph $G$ and then go towards a more general graph $G$. From Lemma D.1,

$$t_{\text{eff},i} \geq t_i + \min\{\rho\Im(i,G), T - t_i\}$$

if $T - t_i \leq \rho\Im(i,G)$, then $t_{\text{eff},i} \geq \frac{T+t_i}{2} \geq \frac{T}{2}$. For the reverse case of $T - t_i \geq \rho\Im(i,G)$, $t_{\text{eff},i} \geq t_i + \frac{\rho\Im(i,G)}{2} \geq t_i + \frac{\rho}{2(n-1)}$ (since $\Im(i,G) \geq \frac{1}{n-1}$ by Remark **??**).

Since $\boldsymbol{\pi}_T$ is a cyclic sampling policy, hence $t_i$ increases by 1 at-least once every $n$ samples. Thus, we can upperbound the infinite sum as,

$$\begin{aligned}
\sum_{T=1}^{\infty} \frac{1}{t_{\text{eff},i}^2} &\leq \sum_{T=1}^{\infty} \frac{1}{\left(t_i + \frac{\rho}{2(n-1)}\right)^2} \\
&\leq n\left(\frac{2(n-1)}{\rho}\right)^2 + n\sum_{t_i=1}^{\infty} \frac{1}{t_i^2} \\
&< n\left(\frac{2(n-1)}{\rho}\right)^2 + \frac{n\pi^2}{6}
\end{aligned} \tag{108}$$

Hence proved. $\qquad\square$

# J  Better sampling strategies

Theorem 4.4 established a baseline w.r.t. sampling protocol by solving $T_{\text{sufficient}}$ for naive cyclic sampling policy (a sampling policy which doesn't exploit the graph properties). Note that, even if the sampling policy doesn't utilize any graph properties, the similarity graph is still being utilized in computing the mean estimate and the confidence widths. For the safe elimination of suboptimal arms, the ultimate goal of GRUB is to shrink the confidence bounds $\beta_i\sqrt{(t_{\text{eff},i})^{-1}}$ as quickly as possible. Accordingly, a few intelligent sampling policies that exploit the graph structure of the problem is given as follows:

- **Marginal variance minimization (MVM):** Since picking any arm impacts the confidence widths of all arms in it's connected component, we pick the arm with the maximum variance. Specifically, $l = \underset{i \in A}{\arg\min}\, t_{\text{eff},i} = \underset{i \in A}{\arg\max}\, [V_T^{-1}]_{ii}$, where $A$ is the set of indices of the arms under consideration.

- **Joint variance minimization – nuclear (JVM-N):** This variant is inspired from the concept of V-optimality [22]. This policy aims to select the arm that minimizes $\ell_2$ regression loss of the estimated vector $\hat{\boldsymbol{\mu}}$, i.e. the confidence interval across all remaining arms in $A$. Specifically, $l = \underset{i \in A}{\arg\min}\|(V_T + \mathbf{e}_i\mathbf{e}_i^T)^{-1}\|_*$

- **Joint variance minimization – operator (JVM-O).** Taking inspiration from $\Sigma$-optimality [40, 38] the next policy can be stated as, $l = \underset{i \in A}{\arg\min}\|(V_T + \mathbf{e}_i\mathbf{e}_i^T)^{-1}\|_{\text{op}} = \underset{i \in A}{\arg\max}\frac{\|\text{Row}_i(V_T^{-1})\|_2^2}{1+[(V_T^{-1})_{ii}]}$

The main objective of sampling policies is to *decrease* the value of $[V_T^{-1}]_{ii}$ for every arm $i$ as fast as possible. The notion of *decrease* leads to different sampling policies for GRUB. The algorithm chooses the arm which maximizes this notion of decreases.

The objective of the sampling policy **Joint variance minimization – operator (JVM-O)** is equivalent to:

$$\max_{k \in A} \sum_{j \in [n]} |(V_T^{-1})_{k,j}| - |\left(V_T + \mathbf{e}_k\mathbf{e}_k^T\right)_{k,j}^{-1}| \tag{109}$$

Using Sherman-morrison rank 1 update we split the summation into different cases:

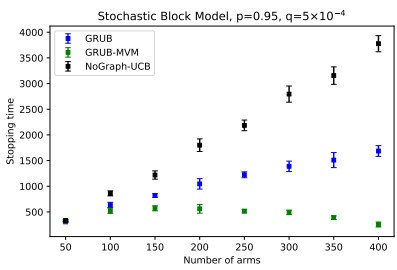
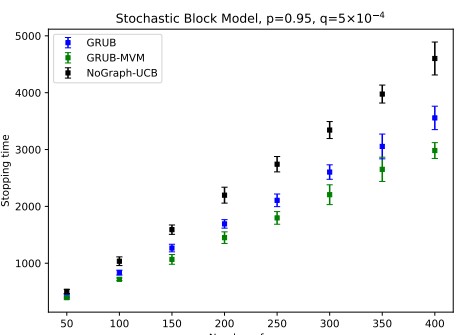

Figure 2: (Best seen in color) Performance of GRUB with using various sampling protocols for SBM $((p, q) = (0.9, 5e-3))$ [Left] and BA $(m = 2)$ [Right]. The UCB method without graph information is significantly slower compared to the graph-based variants. Note that for these toy datasets, the sampling algorithm used does not alter the results too much.

- For $j = k$,

$$|\langle \mathbf{e}_k V_T^{-1} \mathbf{e}_k \rangle| - |\langle \mathbf{e}_k \left( V_T + \mathbf{e}_k \mathbf{e}_k^T \right)^{-1} \mathbf{e}_k \rangle| = \frac{\|\mathbf{e}_k\|_{V_T^{-1}}^4}{1 + \|\mathbf{e}_k\|_{V_T^{-1}}^2} \tag{110}$$

- For all connected-nodes of $j \in \mathcal{N}_k$,

$$|\langle \mathbf{e}_k V_T^{-1} \mathbf{e}_j \rangle| - |\langle \mathbf{e}_k \left( V_T + \mathbf{e}_k \mathbf{e}_k^T \right)^{-1} \mathbf{e}_j \rangle| = \frac{\langle \mathbf{e}_j, \mathbf{e}_k \rangle_{V_T^{-1}}^2}{1 + \|\mathbf{e}_k\|_{V_T^{-1}}^2} \tag{111}$$

- For all other non-connected $j \notin \mathcal{N}_k, i \neq k$,

$$|\langle \mathbf{e}_k V_T^{-1} \mathbf{e}_i \rangle| - |\langle \mathbf{e}_k \left( V_T + \mathbf{e}_k \mathbf{e}_k^T \right)^{-1} \mathbf{e}_i \rangle| = 0 \tag{112}$$

Hence the sampling policy decides on the arm to sample based on the following optimization problem,

$$\sum_{j \in [n]} |(V_T^{-1})_{k,j}| - |\left( V_T + \mathbf{e}_k \mathbf{e}_k^T \right)^{-1}_{k,j}| = \frac{\|\mathbf{e}_k\|_{V_T^{-1}}^4 + \sum_{j \in \mathcal{N}_k} \langle \mathbf{e}_i, \mathbf{e}_k \rangle_{V_T^{-1}}^2}{1 + \|\mathbf{e}_k\|_{V_T^{-1}}^2}$$

$$= \frac{\|\mathrm{Row}_k(V_T^{-1})\|_2^2}{1 + [(V_T^{-1})_{kk}]}$$

Hence we try to find the arm $k$ within the remaining arms in consideration which maximizes $\frac{\|\mathrm{Row}_k(V_T^{-1})\|_2^2}{1 + [(V_T^{-1})_{kk}]}$.

## K  Additional Experiments

**Synthetic Data : Setup 2** We consider an $n$-armed bandit setup with the aim of finding the best arm. The number of arms scale from $n = 50$ to $200$ in steps of $50$. We consider 2 cases: $G$ is a Stochastic Block model(SBM) with parameters $(p, q) = (0.9, 1e^{-4})$ and $G$ is a Barabási–Albert(BA) graph with parameter $m = 2$, both containing 10 clusters. We run every setup for 20 runs and record the stopping time for all runs.

As can be seen in 1, all graph algorithm (GRUB with different sampling policies) the standard UCB based best-arm identification algorithm. Within the different GRUB, different sampling policies exploit the graph infromation in distinct ways, leading to a different in their performance. GRUB (cyclic sampling based) is outperformed by all other sampling based GRUB methods.

**Synthetic Data: Setup 3** We consider the setup where $G$ with $n = 100$ arms consists of 10 connected components with 10 arms per cluster. We consider 2 cases: $G$ is a Stochastic Block model(SBM)

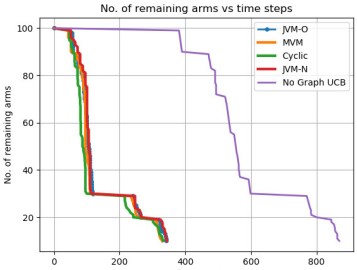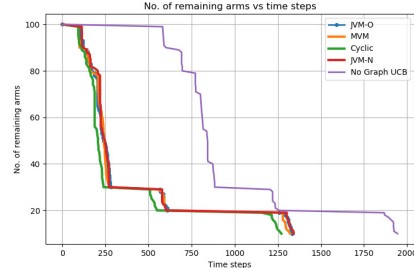

Figure 3: (Best seen in color) Performance of GRUB with using various sampling protocols for SBM $((p, q) = (0.9, 5e - 3))$ [Left] and BA $(m = 2)$ [Right]. The UCB method without graph information is significantly slower compared to the graph-based variants. Note that for these toy datasets, the sampling algorithm used does not alter the results too much.

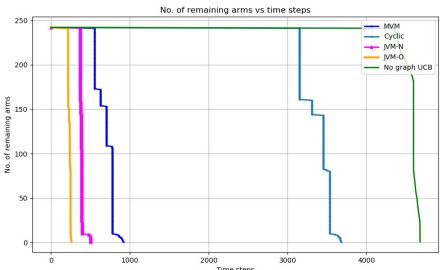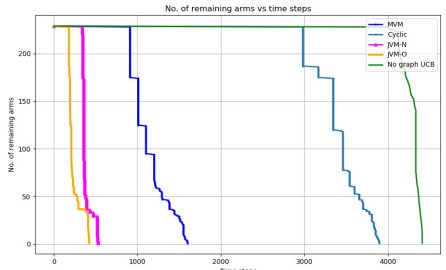

Figure 4: (Best seen in color) Performance of GRUB using different sampling protocols for Github social graph (left) and LastFM graph (right). With no graph information, UCB requires orders of magnitude more samples compared to policies that use explicitly graph information. The cycic sampling policy is not as competitive on real world datasets

with parameters $(p, q) = (0.9, 0.0005)$ and $G$ is a Barabási–Albert(BA) graph with parameter $m = 2$. Results are provided in Figure 3

**Real Data:** We use graphs from SNAP [34] for the experiments involving real world graphs. We sub-sample the graphs using Breadth-First Search (to retain connected components) to generate the graphs for our experiments. We use the LastFM [46], subsampled to 229 nodes and Github Social [45] subsampled to 242 nodes.

In all the experiments, it is evident that GRUB with any of the sampling policies outperform UCB algorithm [32], which does not leverage the graph. Further within the various sampling policies, MVM sampling policy seems to outperform other sampling policies (Figure 4). For both Github and LastFM datasets, the MVM policy obtains the best arm in $\sim 300$ rounds compared to traditional UCB that takes $\sim 4500$ rounds. A rigorous theoretical characterization of the above sampling policies is an exciting avenue for future research. We refer the reader to Appendix A for a discussion on the results of the paper, potential extensions, and broader impacts.

# L   Code Availability

The full code used for conducting experiments can be found at the following Github repository.