# OpenReview forum: "Maximizing and Satisficing in Multi-armed Bandits with Graph Information"
_NeurIPS.cc/2022/Conference — NeurIPS 2022 Accept_

### Official Review · Reviewer_XXzj · 2022-07-13

**Rating:** 7
**Confidence:** 4
**Soundness:** 4 excellent
**Presentation:** 2 fair
**Contribution:** 3 good

**Summary:**

The paper tackles the problem in which a MAB setting with a pure-exploration goal having a large number of arms, has also some structure among the arms by means of a similarity matrix stating how much the expected value of the arms can differ from each other. They propose an algorithm able to exploit such a structure and analyse the problem in terms of lower bound complexity. Finally, they show that exploiting the structure of the graph provides significant improvements over the standard MAB algorithms.

**Questions:**

Could you give more details on the settings in which the smoothness assumption is satisfied? How it is possible to infer \epsilon?

What happens if the algorithm is used on settings in which the assumption about the smoothness of the problem?

Did you compare your approach also with Thompson Sampling?

**Limitations:**

I do not foresee limitations and potential negative societal impacts of this work.

**Strengths And Weaknesses:**

The topic analysed by the paper is interesting. The setting is clear, even if it is not clear how often the assumption required by the formulation of the problem is satisfied in real settings.

The paper requires some proofreading to polish it from orthographic errors. I mentioned a few of them in the minor section. Overall it is clear and well structured.

Overall, the paper is not self-contained. In my opinion, the authors should have done a better job in the identification of the crucial points of the paper, avoiding repetitions and inclusion of the elements which are crucial to having a self-contained paper. For instance, you decided to defer the parameters c_1 and c_2 to the appendix. The same holds for the pseudo-codes of the algorithms and the definitions of the sampling strategies.

Minor:
Line 85 (and following comments) State that "potentially without playing it even once!" is not necessary.
Line 155: "natural" it is not clear to me what you mean with "natural"
Line 159: sub-optima -> sub-optimal
Line 178> utilization -> use
Line 217: e.g -> e.g.,
Line 221: atleast -> at least
Line 279: in mean -> in terms of expected reward

---

> ### Author Response · Authors · 2022-08-02
> **Response to Reviewer XXzj**
>
> We thank the reviewer for their careful reading and insightful comments. We provide specific answers to the questions raised below.
>
> **Q : Could you give more details on the settings in which the smoothness assumption is satisfied? How it is possible to infer $\epsilon$?**
>
> **A :** In our work, we use an *upper-bound* on the side-information error $\epsilon$, rather than an exact value of the constraint violation. Such an upper bound $\epsilon$ can be obtained by multiple ways : either by a domain expert (and is like a regularization parameter in other statistical learning methods), by constructing it using feature vectors, by employing cross-validation or the so-called ``doubling trick'' (where one can run the algorithm with a sequence of exponentially increasing values of $\epsilon$ until acceptable results are obtained; this would only add a logarithmic factor to the sample complexity). Expert knowledge is typically available; for instance, in drug discovery, where knowledge about the chemical similarities of various compounds (*Knowledge graphs and their applications in drug discovery, Finlay 2021*) is available and this can be codified and used with our framework. Other biomedical applications have also been explored by many (*Constructing knowledge graphs and their biomedical applications. Nicholson 2020*) where this could potentially be utilized. Equivalently, domain knowledge might allow the representation of the arms (or actions) in the bandit problem using feature vectors which may in-turn be used to compute similarities and form similarity graphs.
>
> **Q : What happens if the algorithm is used on settings in which the assumption about the smoothness of the problem?**
>
> **A :** As is the case in several learning and inference problems that incorporate structure, the benefit of correct structural information can be immense as we demonstrate in this paper. However, as in these situations, there is a risk of degraded performance when such structural information is misleading. This phenomenon comports with our intuition about the benefit of inductive bias and is at the heart of several ``no free lunch'' theorems. However, the point raised by the reviewer is an important one and we will include a discussion about this in the paper. We think an exciting avenue for future work is extending the proposed setting and algorithms to the case where one has model-misspecification, where the algorithm is required to be robust even in situations where the graph provided is inaccurate or noisy. This is indeed what we view as a natural next step for this work.
>
> **Q : Did you compare your approach also with Thompson Sampling?**
>
> **A :** This is an interesting suggestion! As the reviewers note, this is the first piece of work to the best of our knowledge that rigorously demonstrates the advantage of having such side information for pure exploration problems. The consideration of alternative strategies for tackling this and related problems, such as Thompson sampling, is indeed an exciting avenue for future work.
>
> We also thank the reviewer for their comments on typographical and formatting errors and will be sure to correct them in our revised manuscript.

---

### Official Review · Reviewer_vans · 2022-07-26

**Rating:** 7
**Confidence:** 2
**Soundness:** 4 excellent
**Presentation:** 4 excellent
**Contribution:** 3 good

**Summary:**

In this work, the authors introduce and study a novel variant of the multi-armed bandit problem, where side-information on the similarity (in mean reward) among different arms is provided to the decision-maker. In particular, the authors consider a weighted graph on the arms/nodes, where the weight of each edge indicates how similar the two "adjacent" arms are. For this model, the authors provide a novel algorithm for the problem of best-arm identification, for which they provide associated guarantees. By providing a lower bound, they show that their algorithm achieves minimax optimality for an interesting class of instances. A variant of the algorithm for the problem of \zeta-best-arm identification is also provided. Finally, the authors evaluate the empirical performance of their algorithm on synthetic and real data.


**Questions:**

- Could the authors comment on how the smoothness parameter \epsilon scales (or should naturally scale) with the number of arms? Specifically, consider the example of a (dense) unit-weight random graph, where each edge appears with probability p = 0.1. What is \epsilon in that case? To rephrase, how \epsilon increases by adding a new arm to the graph? How this affects the provided guarantees compared to the case where no similarity structure is assumed?


- Personally, I would move the section on $\zeta$-GRUB to the Appendix (I do not think it contributes much to the main part). This would free some valuable space to add the pseudocode of GRUB and/or maybe some proof sketches of the main results.


- Maybe it would be reasonable to discuss the line of work "Online Learning with Gaussian Payoffs and Side Observations" (NeurIPS '15) by Wu et al. and "Asymptotically-optimal Gaussian bandits with side observations" (ICML'22) by Atsidakou et al.
The side information structure assumed there (weighted graph) bears some similarities with the one considered in this paper.

**Limitations:**

Yes

**Strengths And Weaknesses:**

STRENGTHS:

- Overall, the paper is very well-written and well-organized (including parts of the Appendix).

- The proposed model is (to the best of my knowledge) novel, and applies to several scenarios where the number of actions is very large, yet some form of similarity structure between "adjacent" actions is available (e.g., social networks).

- The proposed algorithm cleverly exploits the given graph structure and achieves minimax optimality a special class of natural problems.

WEAKNESSES:

- Assuming some form of graph-encoded similarity structure is a reasonable assumption (e.g., all friends in a social network can have unit weight). However, knowledge of the smoothness parameter \epsilon on the graph seems critical for the perormance of the algorithm, which, in my opinion, weakens the obtained results.

---

> ### Author Response · Authors · 2022-08-02
> **Response to Reviewer vans**
>
> We thank the reviewer for their careful reading and insightful comments. We provide specific answers to the questions raised below.
>
> **On the knowledge of $\epsilon$**
>
> The goal of this paper is to consider pure exploration problems in multi-armed bandits when there is side information available about the arm similarities and the associated similarity in arm rewards. That is, the side information the algorithm needs is *both* the graph $G$ and (an upper bound on) the constant $\epsilon$. Indeed, given a graph, one can have mean rewards that exhibit both extremely low and extremely high Laplacian norms, and we are interested in the former case as it signals that the graph captures non-trivial similarity information about the arms. Typically such information may be  available from a domain expert or may be computed using feature representations of the arms. The reviewer raises an interesting prospect where information about how smooth the rewards are with respect to the graph is not available. We think this is an important avenue for follow on work where one may employ strategies such as cross-validation ($\epsilon$ may essentially be thought to play the role of a regularization parameter) or using the so-called ``doubling trick'' (where one runs a sequence of instances of the algorithm with exponentially increasing values of $\epsilon$ until acceptable results are obtained; this would only result in a logarithmic inflation in sample complexity). We thank the reviewer for raising this important question and we will include an expanded discussion in our revised manuscript.
>
> **Q : Could the authors comment on how the smoothness parameter $\epsilon$ scales (or should naturally scale) with the number of arms? Specifically, consider the example of a (dense) unit-weight random graph, where each edge appears with probability $p = 0.1$. What is $\epsilon$ in that case? To rephrase, how $\epsilon$ increases by adding a new arm to the graph? How this affects the provided guarantees compared to the case where no similarity structure is assumed?**
>
> **A :** This is an interesting question! As we mentioned in the previous answer, it is worth noting that for *any graph* $G$, one can have a reward vector $\pmb{\mu}$ such that $\langle \pmb{\mu}, L_G \pmb{\mu}\rangle$ can take on a range of values. Indeed, this range is dictated by the spectrum of $L_G$ and one may partially address the reviewer's random graph question based on this. We will include a discussion in our revised manuscript.
>
> Interestingly, inspired by the reviewer's comments, we also derived an upper bound on the value of $\epsilon$ (which essentially constrains how ``non-smooth'' the mean rewards can be with respect to the graph) that will ensure that the graph side information provides a provable advantage over the traditional (graph-free) counterpart of the problem; the upper bound expression we have is: $\min_{j \in [n]/i^\ast}\frac{\mathfrak{I}(j, G)\Delta_j^2}{2\rho}$ ($\rho$ is the regularization parameter, $i^*$ is the best-arm and $\mathfrak{I}$ is the influence factor). Note that the upper bound, as one may expect, depends on the reward vector $\pmb{\mu}$ and the structure of the graph (via $\mathfrak{I}$). We thank the reviewer again for this insightful question and we will add a discussion about this in our revised manuscript.
>
> **Q : Maybe it would be reasonable to discuss the line of work "Online Learning with Gaussian Payoffs and Side Observations" (NeurIPS '15) by Wu et al. and "Asymptotically-optimal Gaussian bandits with side observations" (ICML'22) by Atsidakou et al. The side information structure assumed there (weighted graph) bears some similarities with the one considered in this paper.**
>
> **A :** Thanks for these great pointers! We will include a discussion of these papers in the updated version of our manuscript. It is worth noting that the above papers are under a different setting where each arm play explicitly reveals information about other arms (albeit at a degraded level).

---

### Meta-Review · Area_Chair_xDgL · 2022-09-11

**Recommendation:** Accept
**Confidence:** Certain

**Metareview:**

All reviewers agree on the merits of the work.

**Award:**

No

---

### Decision · Program_Chairs · 2022-09-14

Accept